# Mobile monitoring of urban air quality at high spatial resolution by low-cost sensors: Impacts of COVID-19 pandemic lockdown

Shibao Wang[1], Yun Ma[1], Zhongrui Wang[1], Lei Wang[1], Xuguang Chi[1], Aijun Ding[1], Mingzhi Yao[2], Yunpeng Li[2], Qilin Li[2], Mengxian Wu[3], Ling Zhang[3], Yongle Xiao[3], Yanxu Zhang[1]

[1]School of Atmospheric Sciences, Nanjing University, Nanjing, China
[2]Beijing SPC Environment Protection Tech Company Ltd., Beijing, China
[3]Hebei Saihero Environmental Protection Hi-tech. Company Ltd., Shijiazhuang, China

**Correspondence:** Yanxu Zhang (zhangyx@nju.edu.cn)

**Abstract.** The development of low-cost sensors and novel calibration algorithms provides new hints to complement conventional ground-based observation sites to evaluate the spatial and temporal distribution of pollutants on hyperlocal scales (tens of meters). Here we use sensors deployed on a taxi fleet to explore the air quality in the road network of Nanjing over the course of a year (Oct. 2019–Sep. 2020). Based on GIS technology, we develop a grid analysis method to obtain 50 m resolution maps of major air pollutants (CO, $NO_2$, and $O_3$). Through hotspots identification analysis, we find three main sources of air pollutants including traffic, industrial emissions, and cooking fumes. We find that CO and $NO_2$ concentrations show a pattern: highways > arterial roads > secondary roads > branch roads > residential streets, reflecting traffic volume. While the $O_3$ concentrations in these five road types are in opposite order due to the titration effect of NOx. Combined the mobile measurements and the stationary stations data, we diagnose that the contribution of traffic-related emissions to CO and $NO_2$ are 42.6 % and 26.3 %, respectively. Compared to the pre-COVID period, the concentrations of CO and $NO_2$ during COVID-lockdown period decreased for 44.9 % and 47.1 %, respectively, and the contribution of traffic-related emissions to them both decreased by more than 50 %. With the end of the COVID-lockdown period, traffic emissions and air pollutant concentrations rebounded substantially, indicating that traffic emissions have a crucial impact on the variation of air pollutants levels in urban regions. This research demonstrates the sense power of mobile monitoring for urban air pollution, which provides detailed information for source attribution, accurate traceability, and potential mitigation strategies at urban micro-scale.

## 1 Introduction

Urban air pollution poses a serious health threat with > 80 % of the world's urban residents exposed to air pollution level that exceed the World Health Organization (WHO) guidelines (WHO, 2016). The global urban air pollution (measured by $PM_{10}$ or $PM_{2.5}$) also deteriorated by 8 % during recent years despite improvement in some regions (WHO, 2016). Extremely large spatial variability exists for urban air pollutants [e.g., carbon monoxide (CO), nitrogen dioxide ($NO_2$), and ozone ($O_3$)] over scales from kilometer to meters, as a result of complex flow pattern, non-linear chemical reactions, and unevenly distributed emissions from vehicle and industrial activities (Apte et al., 2017; Miller et al., 2020). Here we illustrate an approach to obtain a high-resolution urban air quality map using low-cost sensors deployed on a routinely operating taxi fleet.

High spatio-temporal resolution air quality data is critical to urban air quality management, exposure assessment, epidemiology study, and environmental equity (Apte et al., 2011, 2017; Boogaard et al., 2010). Numerous methodologies have been developed to obtain urban air pollutant concentrations, including stationary monitoring networks (Cavellin et al., 2016), near-roadway sampling (Karner et al., 2010; Zhu et al., 2009; Padro-Martinez et al., 2012), satellite remote sensing (Laughner et al., 2018; Xu et al., 2019), land use regression (LUR) models (Weissert et al., 2020), and chemical transport models (CTMs) (Li et al., 2010). However, the stationary monitoring stations (including near-roadway sampling) are sparse and uneven, and the ability to reflect the details of urban air pollution is limited, especially at remote communities (Snyder et al., 2013). Remote

sensing and CTMs are generally spatially coarse (~km resolution), and cannot resolve species that are inert to radiative transfer (e.g. mercury and lead) or without known emission inventory and/or chemical mechanisms. LUR model can estimate concentrations at high spatial resolution, but it provides limited temporal information, and the predicting power is poor in areas with specific local sources (Kerckhoffs et al., 2016).

Mobile monitoring is a promising approach to garner high spatial resolution observations representative of the community

scale (Miller et al., 2020; Hasenfratz et al., 2015). Different vehicle platforms are used for mobile monitoring, including minivan (Isakov et al., 2007), cargo tricycle (Airparif, 2009), bicycle (Bart et al., 2012), taxi (O'Keeffe et al., 2019), Street View cars (Apte et al., 2017), and city bus (Kaivonen and Ngai, 2020). However, the scale of deployment and subsequent data coverage are limited by the cost of the observation instrument (Bossche et al., 2015). This issue has been addressed by the development of low-cost sensors that are calibrated with machine learning based algorithms (Miskell et al., 2018; Shiva et al.,

2019; Lim et al., 2019). The emergence of low-cost air monitoring technologies was recognized by the U.S. EPA (Snyder et al., 2013) and European Commission (Kaur et al., 2007), and was also recommended to be incorporated in the next Air Quality Directive (Borrego et al., 2015). For example, Weissert et al. (2020) combined land use information with low-cost sensors to obtain hourly $O_3$ and $NO_2$ concentrations distribution at a resolution of 50 m. High agreements were also found between well-calibrated low-cost sensor systems and standard instrumentations (Chatzidiakou et al., 2019; Hagan et al., 2019).

The objective of this study is to illustrate the sensing power of low-cost sensors deployed on a routinely operating taxi fleet platform in a megacity. We combine mobile observations and geographic information system (GIS) to obtain the hourly distribution of multiple air pollutant concentrations at 50 m resolution. By comparing to the measurements by background sites, the contribution of traffic emission to urban air pollution is also diagnosed. We explore the influencing factors of pollutant levels including time of the day/week and holidays. Moreover, our sampling period covered the outbreak of COVID-19 in

China. The pandemic lockdown had a tremendous impact on the socio-economic activities especially the traffic sector, and subsequently the air quality (Zhang et al., 2020; Huang et al., 2020). We evaluate how urban air quality changes at different periods of the pandemic and explore the impact of traffic-related emissions.

## 2 Materials and methods

### 2.1 Mobile monitoring

We use the mobile sampler XHAQSN-508 from Hebei Sailhero Environmental Protection High-tech Co., Ltd. (Hebei, China) to measure the air quality in Nanjing urban area. The instrument is equipped with internal gas sensors for CO (model XH-CO-50-7), $NO_2$ (XH-NO2-5AOF-7), and $O_3$ (XH-O3-1-7) (dimensions: 290 ×81 ×55 mm; weight: 1.0 kg) as well as two small in-built sensors for temperature and relative humidity, and is fixed in the top lamp support pole (~1.5 m above ground) of two Nanjing taxis (Fig. 1). Two taxis fueled with electricity and liquefied natural gas (one each) are selected to reduce the

impact of emissions from the sampling vehicles themselves. All three sensors are electrochemical, which based on a chemical reaction between gases in the air and the electrode in a liquid inside a sensor that can detect gaseous pollutants at levels as low as ppb (Maag et al., 2018). Sensors are continuously powered by an external DC 12 V power supply provided by a taxi battery. The sample is refreshed by pumping air to the sensors. There is an air inlet at the bottom of the instrument, which is also checked periodically to avoid blockage. Because it is fixed in the taxi top lamp, it can reduce the impact of different wind

direction airflow. This device integrates components for data integration, processing, and transmission, and provides data management, quality control, and visualization functions. Pollutant concentration data is generated by different voltage values sensed by gas sensors, which is automatically uploaded to a database in the cloud via the 4G telecommunications network. We continuously measured the concentration of CO, $NO_2$, and $O_3$ in the street canyon in the urban area of Nanjing (with the center located at 32.07 °N and 118.72 °E) for a whole year (Oct 1, 2019–Sep 30, 2020). An instantaneous measurement of CO, $NO_2$,

and $O_3$ concentrations is configured to continuous measure at a frequency of once per 10 s sampling interval, and their limit of detection (LOD) are 0.01 μmol mol$^{-1}$, 0.1 nmol mol$^{-1}$, and 0.1 nmol mol$^{-1}$, respectively. The sampling routes were relatively random during taxi operations, mainly on the arterial roads. A GPS device (U-blox, Switzerland) is utilized to record the spatial coordinates and the spatial offsets are corrected by Arcgis 10.2 software. Generally, the sampling campaign is conducted on both weekdays and weekends from 6:00 A.M. to 10:00 P.M. Occasionally the taxi drivers work for the night shift, and the 85 instruments are run from 10:00 P.M. to 6:00 A.M. The collected data covers 373 km$^2$ with a population of 6 million (Fig. 1).

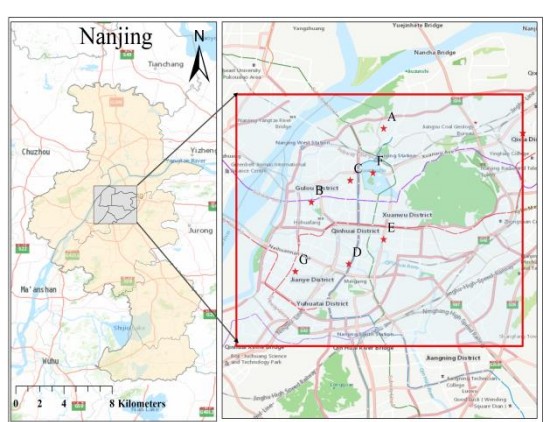
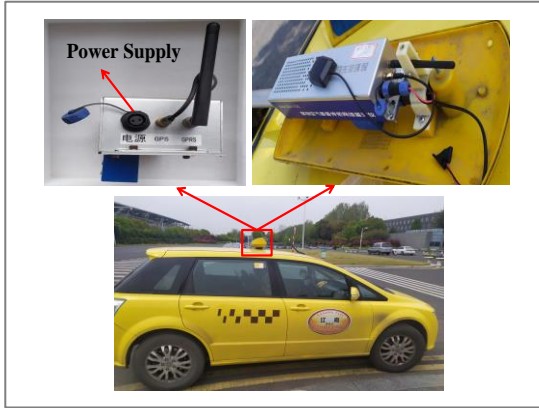

**Figure 1.** Location of the monitoring areas in the city of Nanjing (left) and photo of instrument installment (right). Red stars are the locations of stationary stations belonging to the national air quality measurement network of China. These stations cover different functional regions of the city: A, B, C, D, E, F, and G represent industrial, cultural and educational, commercial, traffic, residential, 90 ecological park and new urban area, respectively. Map credit: ESRI 2020.

### 2.2 Sensors calibration and validation

Different from traditional instruments, low-cost sensors have some limitations, such as nonlinear response, signal drift, environmental dependencies, low selectivity, and cross-sensitivity, so it is important that calibration procedures are applied with respect to these limitations (Maag et al, 2018; Lösch et al., 2008). For example, environmental conditions are known to 95 cause nonlinear behavior of sensors (Popoola et al., 2016). Due to aging and impurity effects over a long time, low-cost sensors are prone to signal drift and low sensitivity (Kizel et al., 2018). In addition, cross-sensitivities differ largely according to the ambient temperature and level of gas the sensor is being exposed to (Lösch et al., 2008). So, multi-parameter joint calibration training is utilized to reduce the interference issue between air pollutants in our research, including air pollutant concentrations, temperature and relative humidity. The sensors are usually trained with co-located data collected by reference methods before 100 being deployed to actual measuring campaigns (Kaivonen and Ngai, 2020; Chatzidiakou et al., 2019; Bossche et al., 2015).

The XHAQSN-508 is calibrated every month starting from September, 2019. The instrument is placed at the outdoor Station for Observing Regional Processes of the EarthSystem (SORPES) in the Xianlin Campus of Nanjing University (https://as.nju.edu.cn/as_en/obsplatform/list.htm) for at least seven days before the taxi began sampling. The collected data is calibrated against standard instruments (Thermo Fisher Scientific 48i, 42i, and 49i, USA for CO, $NO_2$, and $O_3$, respectively). 105 The instrument precision is ±2 ppbv for $O_3$, and ±1 % and ±4 % for CO and $NO_2$, respectively, which have been used in many other studies and found to perform well for long-term runs (Ding et al., 2013; Herrmann et al., 2013). One drawback of our study is that the air pollutant concentrations observed at SORPES are lower than those observed in a road environment, which might cause issues for the calibration process. Comparing different calibration models, we found that machine learning algorithm can improve sensor/monitor agreement with reference monitors, and many previous studies have used this method 110 (Qin et al., 2020; Esposito et al., 2018; Vito et al., 2018). A supervised machine learning methodology based on the Gradient Boost Decision Tree (GBDT) is used for data calibration (Johnson et al., 2018). GBRT, an ensemble learning method, is a decision tree-based regression model that implements boosting to improve model performance using both parameter selection and k-fold cross validation. GBRT needs to be trained by the dataset with target labels (Yang et al., 2017). It takes input variables including raw signals of sensors, air pollutant concentrations (CO, $NO_2$, and $O_3$), temperature and relative humidity.

The stationary instrument data are taken as training targets. The parameters of the machine learning model are adjusted continuously based on gradient descent algorithm. The $R^2$ of the calibration results are generally high ($> 0.90$) for all the three air pollutants (e.g. Fig. 2a).

The success of supervised model training with target labels (i.e. co-located with SORPES, Fig. 2a) does not guarantee for its predicting power for conditions without labels (i.e. on road or co-located with SORPES but not feeding the station data to the algorithm, Fig. 2b). We use a calibration-validation methodology to evaluate the performance of the calibrated sensors (Chatzidiakou et al., 2019). This method includes two phases: first, the sampler was calibrated against the SORPES station for 10 days (Jun. 1-10, 2020), and the sensor data were used for sensor algorithm training as above described (Fig. 2a); second, we continued to place the sampler in the station (Jun. 11–17, 2020). However, the sensor data are not used for calibration but directly fed in the algorithm trained in the first phase. The results are compared with the station data (i.e. validation phase, Fig. 2b). We find that the sensor data agree well with standard instrumentation in the second phase. The sensor retrieved CO, $NO_2$, and $O_3$ concentrations are $0.58 \pm 0.12$ mg m$^{-3}$, $8.40 \pm 4.30$ μg m$^{-3}$, $27.3 \pm 16.5$ μg m$^{-3}$ respectively, not significantly different from that by standard instruments ($0.50 \pm 0.10$ mg m$^{-3}$ and $10.5 \pm 6.31$ μg m$^{-3}$, and $32.4 \pm 20.2$ μg m$^{-3}$) ($\alpha = 0.05$, ANOVA analysis). The $R^2$ values remain generally high (0.82–0.97) for different air pollutants (CO and $O_3$) except $NO_2$ ($R^2 = 0.67$). The lower $R^2$ value for $NO_2$ may be associated with the higher humidity during the validation period (Jun. 13–16, 2020). As $NO_2$ is water dissolvable, high relative humidity may lead to a low bias for sensors (Wei et al., 2018). To improve performance of the $NO_2$ model, temperature and relative humidity have also been involved in the training algorithm. However, the interaction between $O_3$ and $NO_2$ may influence the detection accuracy of these two chemicals, especially for $NO_2$ (Ivanovskaya et al., 2001). The accuracy of the sensor generally decreases with time (aka aging) due to the evaporation of the electrolyte (Ribet et al., 2018). However, we find no significant decrease in the $R^2$ values for the three pollutants during our campaign. It seems that the machine-learning algorithm could successfully compensate the aging of the sensors. Field calibration of low-cost sensors is still a challenging task, as it is greatly affected by atmospheric composition and meteorological conditions (Spinelle et al., 2017; Castell et al. 2017). Our results have high $R^2$ values compared to previous studies, indicating relatively high accuracy (e.g. Castell et al. 2017). The results from the two sensors also agree with each other reasonably well, with $R^2$ values ranged 0.97–0.99 for a linear regression. Their data are thus combined in the following analysis to achieve a maximum data coverage. Overall, the sensor results have substantial uncertainty compared to reference methods, we thus focus on the relative temporal and spatial distributions rather than the absolute concentrations.

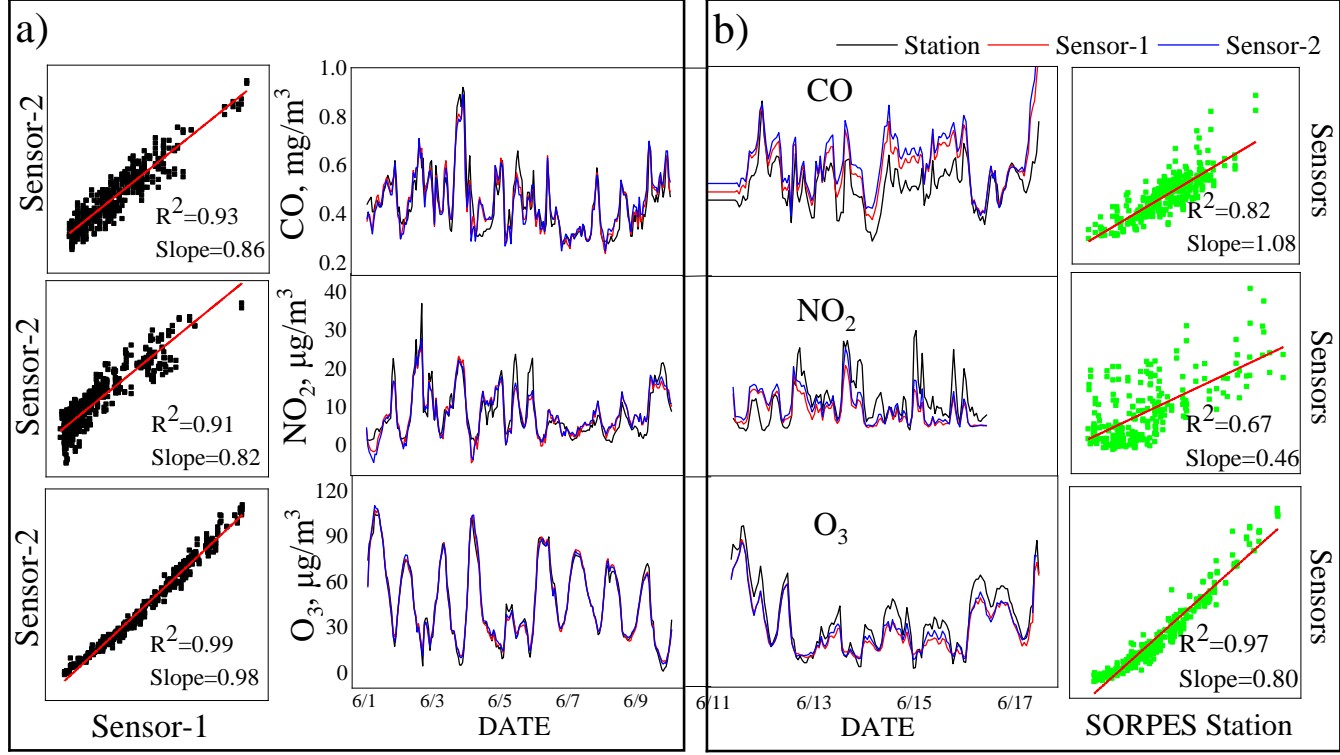

**Figure 2.** Sensor performance evaluated by a calibration-validation methodology for CO, $NO_2$, and $O_3$. a) calibration period (Jun 1–10, 2020); b) validation period (Jun 11–17, 2020). The time series plots compare the concentrations measured by the co-located sensors and standard instruments, while the scatterplots show pollutants concentrations and linear regressions between them.

## 2.3 Data processing

As the mobile monitoring platform samples along the trajectories of carrying vehicles, we need to either sacrifice the temporal information to calculate the spatial distribution of air pollutant, or the spatial information to temporal variations. Similar approaches have also been adopted by previous studies (Bossche et al., 2015; Apte et al., 2017; Farrell et al. 2015). To generate the spatial distribution of air pollutant at high spatial resolution, we divide the research area into grids with 50 m ×50 m resolution, and calculate the mean values of the samples collected in each grid. The driving condition is highly variable and the taxi can travel more than 50 m in 10 seconds if the vehicle speed is over 18 km/hr. However, given the complexity of the driving conditions, we ignore the vehicle trajectory in the past 10 seconds but assign the measured values to the location of the vehicle at the time of data uploading. Then, combined with GIS technology, we calculate the average of all the data points over one year that fall in the same grid. One drawback of our study is the impact of spike concentrations on sensor performance. The sensors keep reporting high concentrations in an approximate one–minute period after exposure to large environmental concentration spikes. This effect would reduce the effective resolution of our gridded concentration map. Similarly, we calculate the hourly average concentrations by considering only the data sampled in the same hour of different days. The GPS signal is missing when the taxis pass through the nine underground tunnels in Nanjing (e.g. Xuanwu lake tunnel and Jiuhuashan tunnel in the city center, Fig. 3). We assume the taxies travel in a constant speed and the sampling points are uniformly allocated along the tunnels. We use the Arcgis 10.2 software for data processing. To calculate the air pollutant concentrations (CO, $NO_2$, and $O_3$) map of different road types and the contribution of traffic emissions to them, we divide the urban roads in Nanjing area into six types, including highways, arterial roads, secondary roads, branch roads, residential streets, and tunnels (https://wiki.openstreetmap.org/wiki/Key:highway). Roads and land use data of Nanjing shown in Fig. 3 are based on OpenStreetMap (OpenStreetMap contributors, 2020).

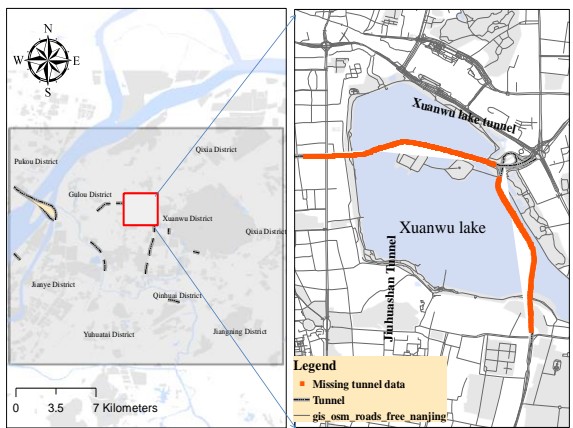

**Figure 3.** Locations of tunnels in Nanjing urban area. © OpenStreetMap contributors 2019. Distributed under a Creative Commons BY-SA License.

## 2.4 Traffic source attribution

The mobile platform keeps sampling in the urban road network which carries a strong signal from traffic sources. By contrast, stationary stations are often located far away from major roads to represent a regional background air pollution level (Hilker et al., 2019). Seven state-operated air quality observation stations in Nanjing are selected in our research, including Maigaoqiao, Caochangmen, Shanxi Road, Zhonghuamen, Ruijin Road, Xuanwu Lake, and Olympic Sports Center (Zhao et a., 2015; Zou et al., 2017), which are far away from major roads and large point sources, so they are usually used as regional backgrounds in different functional areas (Zou et al., 2017; An et al., 2015). For example, Zou et al. (2017) chose the Olympic Center station (G, Fig. 1) to get the background characteristics of CO and $NO_2$ in Nanjing. Therefore, the normalized contribution from traffic-related emissions can be obtained by differencing the mobile measurements and the stationary ones to minimize the influence of daily meteorological variations in the urban air quality, following Bossche et al. (2015):

$$AP_{traffic,ij} = (AP_{ij} - AP_{min})/AP_{ij} \tag{1}$$

where, $AP_{traffic,ij}$ represents the air pollutant concentration contributed by traffic emissions for the i[th] pollutant at time j, %; $AP_{ij}$ is the sensor measured concentration of air pollutant; and $AP_{min}$ means the ambient background concentration, which is calculated as the minimum of the measurements from all the stations in Nanjing in the national air quality network without major sources within a direct vicinity of 50 m (https://quotsoft.net/air/, Fig. 1). We refer to this method as "background site (BS)".

We also adopt a method similar to Apte et al. (2017) for traffic source attribution. This method includes a peak detection algorithm to calculate the contribution of local traffic emission sources to on-road pollutant concentrations. We calculate the mean and minimum of air pollutant concentrations in each grid as the "peak" and "baseline", respectively. The difference between the two is considered as the contribution from traffic sources. We refer to this method as "peak detection (PD)". Matlab R2019a is used for such data calculation.

## 3 Results and discussion

### 3.1 Effect of spatial resolution on reproducibility

There is a trade-off between the resolution of air pollutant concentrations map and its reproducibility, i.e. high-resolution maps subject to large randomness due to the limited number of samples in each grid. We investigate the consistency of spatial patterns of different resolution (10–100 m). We calculate the standard error of the means of samples in each grid (SEM), and then averaged the SEM over all grid cells:

$$SEM = \overline{\sigma/\sqrt{n}} \tag{2}$$

where, σ and n are the standard deviation and number of samples in each grid, respectively. We find the calculated SEM first decays rapidly with the grid spacing but tends to be in a regime of linear decay after a resolution of approximately 50 m for all the three air pollutants (Fig. 4). Therefore, we choose a resolution of 50 m, which is consistent with previous studies (Bossche et al. 2015; Apte et al. 2017). For example, Bossche et al. (2015) used a spatial resolution of 20–50 m to map urban air quality and identify hotspots. Apte et al. (2017) found that reproducible results (with high precision and low bias) of NO, $NO_2$, and black carbon can be generated by at least 10-25 repetitions in a specific area with 30 m median spatial aggregation.

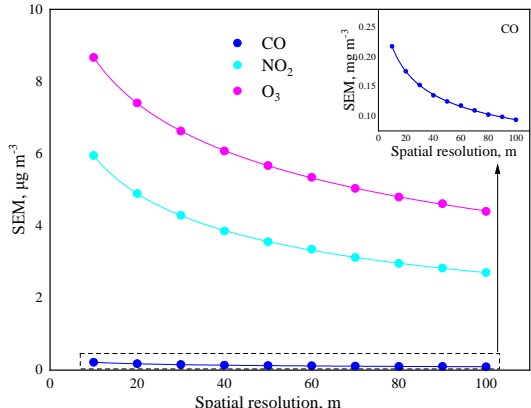

**Figure 4.** Relationship between grid resolution and the domain-averaged standard error of the mean of samples in each grid (SEM) for CO, $NO_2$, and $O_3$.

### 3.2 Road network coverage

A total of 1.32 million pieces of data were obtained during the observation period, which covers 66.4 % of the major roads in Nanjing in the sampling domain with a large repeat-visit frequency [median repetition = 61 (14 and 264 as the lower and upper quartiles, respectively, the same hereinafter)] (Fig. 5a). The type of road with the most visits is the Neihuan lines [258 (116, 526)], followed by the arterial roads [125 (35, 393)], secondary roads [151 (24, 442)], and highways [34 (12, 115)]. The residential streets [22 (6, 100)] have the least visits.

Apart from the areas without roads, such as the Yangtze River, Xuanwu Lake, and Purple Mountain, the data covers 43.5 % of the 50 m grids in the research area with the two taxis contributing 36.8 % and 37.2 %. As shown in Fig. 5b, the median number of repeated frequency in each grid is 66 (18, 286), with the highest value of 15449 in Nanjing South Railway Station and the lowest in some residential roads (1). The repeated frequencies in each 50 m grid along the arterial roads and Neihuan line are higher than other types of roads, i.e. Zhongyang road, Huju road, Neihuandong and Neihuanxi lines (Fig. 5b). Our repeated frequency is generally higher than previous research on mobile monitoring of urban air pollution (Peters et al., 2013; Poppel et al., 2013; Bossche et al., 2015; Apte at al., 2017), which can lower the uncertainty of our results. By comparing the time series of the air pollutant concentrations with that from nearby state-operated air quality observation stations (A' and E', with repeated frequencies > 500), we find that the results are consistent (Fig. S1), which shows the stability and reliability of our data.

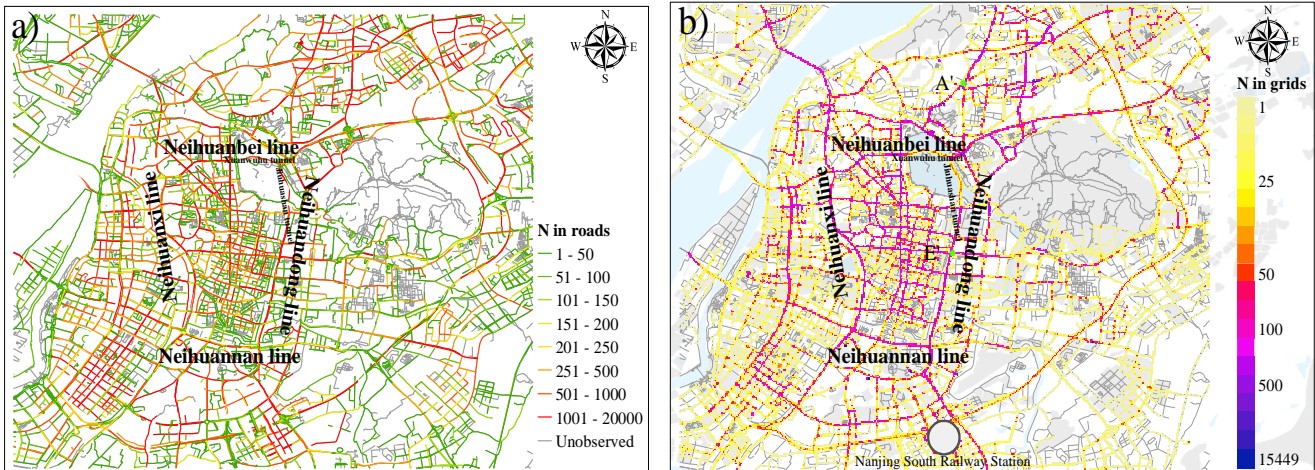

**Figure 5.** Mobile monitoring data coverage with regard to roads (a) and 50 m grids (b). © OpenStreetMap contributors 2019. Distributed under a Creative Commons BY-SA License.

### 3.3 Variability analysis

Fig. 6 and Fig. S2 show the coefficients of variation (CV ≡ standard deviation/ mean × 100 %) for different air pollutants in each grid. For one thing, this matric quantifies the sensing power of mobile monitoring, i.e. more data points reduce uncertainty of observations. For another, it reflects the inherent variability of pollutants caused by factors such as meteorological conditions and hotspots emission sources. We find that the CV values are lower than 100 % on the main roads, including highways and arterial roads, but higher than 100 % on some tunnels, residential streets and Nanjing railway station. As discussed above, the road network coverage is much higher over the main roads than smaller roads. This indicates that increasing the sampling numbers within secondary and residential roads is the most useful to reduce the uncertainty of mobile observation. It is also interesting to notice that a single taxi has a data coverage of ~37 % but the second one only increases it by ~6.5 % to 43.5 %, which implies that the marginal increase of spatial coverage decreases substantially with increasing number of sensors. This is indeed one limitation of our monitoring platform, and much larger fleet size or different sampling platforms (e.g. bikes) may be needed to reduce the uncertainty over these smaller roads.

Although the spatial patterns of CV are similar for different air pollutants, we find generally higher CV for $O_3$ (67.3 %) and $NO_2$ (59.5 %) than CO (51.6 %). This is associated with the spatial and temporal variability of different air pollutants, which are influenced by their lifetimes in the atmosphere. Lifetime (or residence time) is the average time for a chemical compound that is transported in the atmosphere before it is deposited or consumed by chemical reactions. It is associated with its spatial scale of variability. The longer the lifetime, the more uniform the concentrations are distributed. The chemical properties of CO are the most stable in the environment ($\tau$ = 1~2 months), and its spatial concentration difference is more affected by the sampling time and the number of samples. The lifetime of NOx is shorter ($\tau$ = 2~11 hours, Romer et al., 2016), so the measured concentrations are more influenced by local or "hotspot" emissions and meteorological factors. $O_3$ has the shortest lifetime ($\tau$ = ~1 hour in urban atmosphere, McClurkin et al., 2013) among the three pollutants. The level of ozone is affected by its precursors (NOx and VOCs), which both have large variability (Sharma et al., 2016). The complex chemical reactions also increase its spatial heterogeneity.

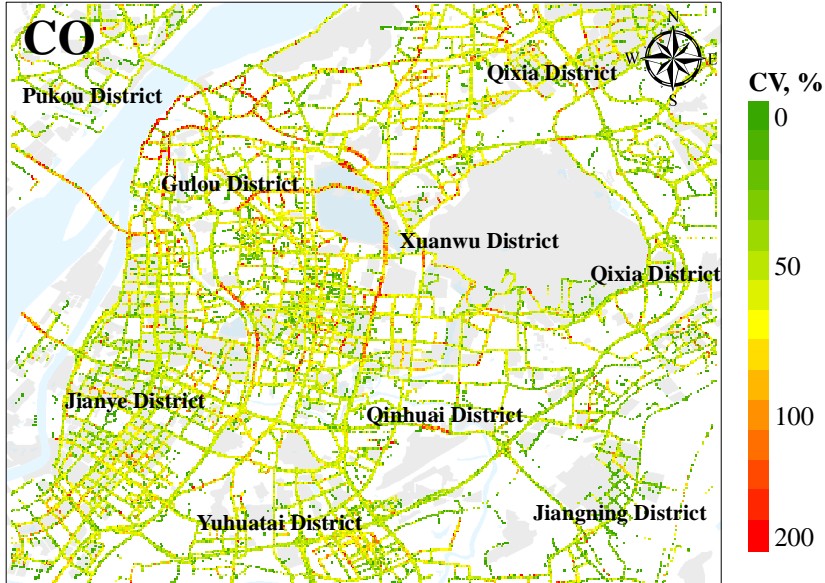

**Figure 6.** Spatial distribution of coefficient of variation for CO in 50 m grids in research domain. © OpenStreetMap contributors 2019. Distributed under a Creative Commons BY-SA License.

## 3.4 Spatial distribution

### 3.4.1 Hotspots identification

Although the instantaneous pollution level varies drastically in different road environments, we obtain a relatively robust time integrated pollution estimate by calculating the mean of repeatedly samples (Fig. 7). We define the area where the pollutant concentrations are 50 % higher than nearby grids (radius = 300 m) as "hotspots" following Apte et al., (2017). The pollutant concentrations shown in Table 1 are the values after deducting the background concentration, which are calculated by the annual mean concentration of stationary stations. A total of 17 hotspots for CO and $NO_2$, and 17 hotspots for $O_3$ are identified, and the specific information is shown in Fig. 7 and Table 1. Most of the "hotspots" show relatively apparent spatial "peaks" for multiple pollutants. To identify the main sources contributing to these hotspots, we use the different relative concentrations of the measured pollutants (Zhao et al., 2015). We also use field information around hotspots area, such as the existence of subway stations, construction sites, factories, and restaurants nearby. This method has substantial uncertainties to attribute the potential sources to these "hotspots", and further source-receptor relationship and detailed chemical component analyze are required to identify the exact emission sources.

We find that "hotspots" are mainly affected by one of the three types of emission sources, namely traffic emissions (diesel and gasoline on-road vehicle exhaust), industrial emissions, and cooking fumes. The mean CO and $NO_2$ concentrations are relatively high at the crossroads (E, 1.47 mg m$^{-3}$ and 15.8 µg m$^{-3}$), tunnels (B, 1.24 mg m$^{-3}$ and 16.6 µg m$^{-3}$, respectively), the roads near the hospital (H, 0.66 mg m$^{-3}$ and 15.7 µg m$^{-3}$), and near the railway station (A, 0.60 mg m$^{-3}$ and 4.0 µg m$^{-3}$), which are affected by on-road traffic emissions. In addition, due to the construction of Maigaoqiao subway station (G, 0.91 mg m$^{-3}$ and 11.8 µg m$^{-3}$), diesel vehicles and off-road traffic emission also make a great contribution to CO and $NO_2$ concentrations. Industrial emissions from petrochemical enterprises (I) also lead to high $NO_2$ concentrations (0.26–93.1 µg m$^{-3}$) on surrounding roads.

As shown in Fig. 7, the higher $O_3$ concentrations in these hotspots area are mainly caused by higher NOx and VOCs emissions from the heavy traffic (W, 46.8±27.4 µg m$^{-3}$, Xie et al., 2016; Ding et al., 2013), cooking emissions (Q, 38.5±26.0 µg m$^{-3}$), and ozone precursors from industrial emissions [e.g., K (47.1±36.5 µg m$^{-3}$) and J (37.6±25.8 µg m$^{-3}$)], such as VOCs. In addition, biogenic VOC emissions also have a significant impact on the formation of ozone [(U (40.4±18.3 µg m$^{-3}$) and V (33.5±20.4 µg m$^{-3}$), Liu et al., 2018)]. Taxi sensor data also reveals the secondary pollution characteristics in micro-

scale, showing that $O_3$ concentration in the downtown area with dense buildings is significantly higher than that in other areas, especially some residential areas in Jianye and Gulou district. Previous studies have also found that the air pollutants "hotspots" are associated with traffic-related emissions [e.g., heavy–duty diesel vehicles (Targino et al., 2016) and vehicle congestion (Gately et al., 2017)] and high-density urban areas (Li et al., 2018). These identified air pollution "hotspots" and the diagnosed source contributions provide helpful information for urban air quality management, which demonstrates the sensing power of mobile monitoring deployed on taxi fleet.

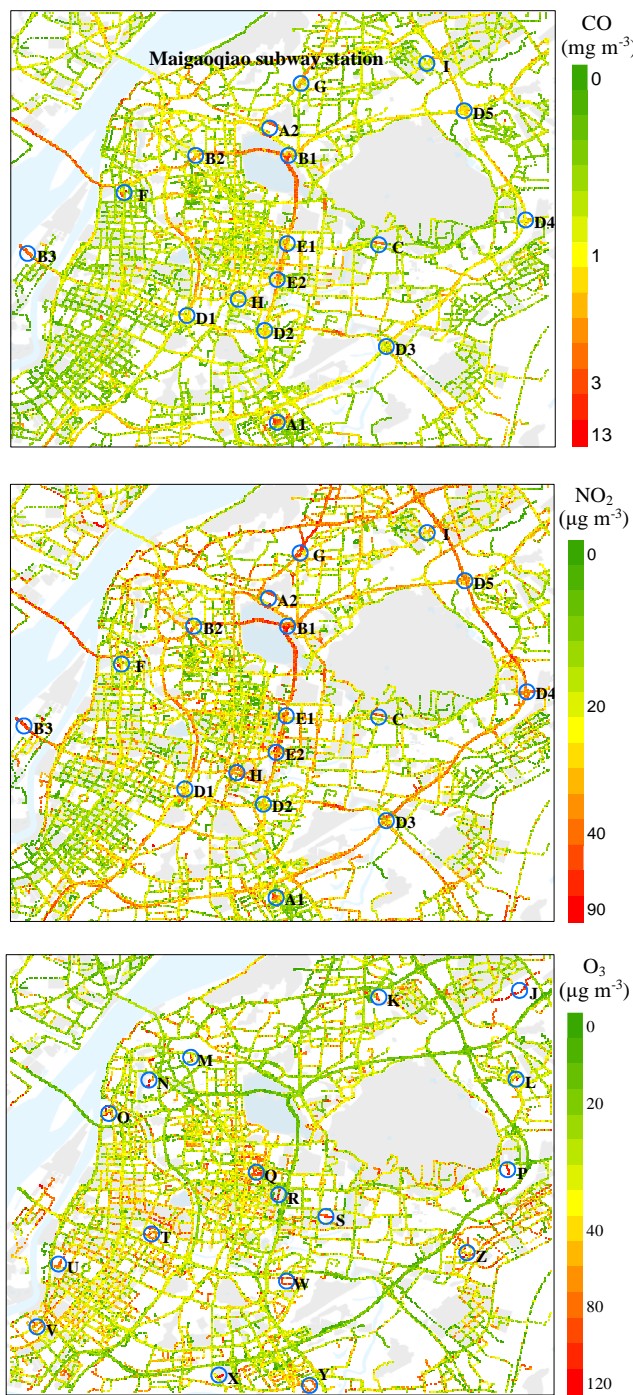

**Figure 7.** Spatial distribution and "hotspots" of air pollutant concentrations in the research domain (CO, NO₂, and O₃). Circles marked with A-Z represent the identified "hotspots", where the air pollutant concentrations are at least 50% higher than the surrounding area (300 m radius). © OpenStreetMap contributors 2019. Distributed under a Creative Commons BY-SA License.

**Table 1.** "Hotspots" of air pollution for multi-pollutants identified in Nanjing.

| ID | Specific | No | CO, mg m$^{-3}$ | NO$_2$, μg m$^{-3}$ | Description/Potential sources |
|---|---|---|---|---|---|
| A | A1, A2 | 6535 | 0.60±0.82 | 4.0±15.9 | Nanjing railway station, gasoline vehicle emission |
| B | B1, B2, B3 | 4177 | 1.24±1.74 | 16.6±26.1 | Exit and entrance of tunnel, gasoline vehicle emission |
| C | C | 1002 | 0.73±0.39 | 0.90±12.5 | Subway entrance, gasoline vehicle emission |
| D | D1–D5 | 4333 | 0.46±0.61 | 6.10±15.0 | Overpass on ring road, vehicle emission |
| E | E1, E2 | 5354 | 1.47±3.04 | 15.8±26.8 | Crossroads, vehicle emission |
| F | F | 1052 | 0.55±0.53 | 13.5±14.2 | Moonlight Plaza/ vehicle emission |
| G | G | 6160 | 0.91±1.31 | 11.8±21.0 | Maigaoqiao subway station, diesel vehicle emission |
| H | H | 6231 | 0.66±0.74 | 15.7±23.5 | Hospital, vehicle emission |
| I | I | 2386 | 0.36±0.49 | 5.60±14.0 | Petrochemical enterprises, Industrial emissions |

No: Observation points within 300 m near the hotspots.

### 3.4.2 Air pollutant concentrations in different types of roads

We find that air pollutant levels differ vastly among the six types of roads ($p < 0.05$, with ANOVA method). The mean CO and NO$_2$ concentrations follow this trend: tunnels (2.22±1.18 mg m$^{-3}$ and 40.7±29.7 μg m$^{-3}$, respectively) > highways (1.10±0.59 mg m$^{-3}$ and 29.2±8.66 μg m$^{-3}$) > arterial roads (0.958±0.308 mg m$^{-3}$ and 25.0±6.90 μg m$^{-3}$) > secondary roads (0.855±0.401 mg m$^{-3}$ and 21.8±8.89 μg m$^{-3}$) > branch roads (0.818±0.216 mg m$^{-3}$ and 20.3±6.79 μg m$^{-3}$) > residential streets (0.783±0.299 mg m$^{-3}$ and 19.7±8.35 μg m$^{-3}$) (Table 2). However, the mean O$_3$ concentrations in different types of roads are opposite to that of CO and NO$_2$: residential streets (35.1±15.4 μg m$^{-3}$) > branch roads (32.7±12.2 μg m$^{-3}$) > secondary roads (31.9±10.0 μg m$^{-3}$) > arterial roads (29.6±7.52 μg m$^{-3}$) > highways (23.3±9.12 μg m$^{-3}$) > tunnels (15.7±7.85 μg m$^{-3}$).

The differences of air pollutant concentrations among different road types are firstly affected by the traffic-related emission sources including vehicle engine exhaust, which is a function of traffic flow and speed, vehicle type, etc. (Sahanavin et al., 2018). The general decreasing trends we observed for CO and NO$_2$ are consistent with traffic flow and congestion index in Nanjing urban area (Table 2, Zou et al., 2017). Apte et al. (2017) also found that the NO$_2$ concentration decreased in turn on highways, arterial roads and residential streets, which are in good agreement with our research. The observed O$_3$ concentrations have opposite trends of CO and NO$_2$ with highest concentration in residential streets (Table 2). As O$_3$ production in Nanjing is in VOC-limited regions, lower NOx could reduce its titration of O$_3$ and subsequently increase O$_3$ concentration (Ding et al., 2013; Xie et al., 2016). The O$_3$ concentrations are lowest in tunnels, which is associated with the weak sunlight in the tunnels (Awang et al., 2015). Furthermore, due to the unfavorable diffusion conditions in the tunnels, NO$_2$ concentration is accumulated to a relatively high level (40.7±29.7 μg m$^{-3}$), which titrates O$_3$. The tunnel also blocks the replenish of surrounding O$_3$-rich air, resulting in lower O$_3$ concentration than other roads (Kirchstetter et al., 1996).

**Table 2.** Multi-pollutant concentrations in six types of roads.

| Road types | Road numbers | Vehicle speed, km/h | Traffic congestion index [a] | CO, mg m$^{-3}$ | NO$_2$, μg m$^{-3}$ | O$_3$, μg m$^{-3}$ |
|---|---|---|---|---|---|---|
| Tunnels | 9 | – | – | 2.22±1.18 | 40.7±29.7 | 15.7±7.85 |
| Highways | 168 | 60~80 | 2.18 | 1.10±0.594 | 29.2±8.66 | 23.3±9.12 |
| Arterials | 443 | 40~60 | 1.78 | 0.958±0.309 | 25.0±6.90 | 29.7±7.53 |
| Secondary | 419 | 30~50 | 1.70 | 0.855±0.401 | 21.8±8.89 | 31.9±10.0 |
| Branch roads | 349 | 20~40 | – | 0.818±0.216 | 20.3±6.79 | 32.7±12.2 |
| Residential | 152 | < 20 | – | 0.783±0.230 | 19.6±8.35 | 35.1±15.5 |

a: The traffic congestion index data is from Gaud map https://report.amap.com/detail.do?city=320100.

**3.5 Temporal variation**

Fig. 8 shows the temporal variation of the three air pollutants concentrations during the observation campaign, with the hourly mean concentrations over the research domain shown in Fig. 9 (the corresponding spatial distributions are shown in Figs. S4–6). The difference of the hourly variation of the mean sample of different types of roads over a year is small (Fig. S7), so the data in Fig. 9 is not filtered in anyway, but for each hour have a similar mix of road types sampled. We find that the median concentrations of CO and $NO_2$ in rush hours (7–9 A.M and 5–7 P.M) are increased by 26.4 % and 27.3 %

compared to non–rush hours, respectively. The hourly mean concentrations of CO and $NO_2$ show a double-peak pattern with higher concentrations in rush hours (Fig. 9a), reflecting the contribution of traffic-related emissions (Tan et al., 2009), which we will elaborate in next section. The observed $O_3$ concentrations show a unimodal diurnal pattern with a peak at ~2 P.M as a result of photochemical formation. At night, $O_3$ concentrations are maintained at a low level due to no solar radiation and NOx–titration effect (Xie et al., 2016; Li et al., 2013). These patterns generally agree with the measurements at stationary

monitoring stations (Fig. S3).

    No significant differences are observed for the median concentrations and spatial distribution of three air pollutants between weekdays and weekends ($\alpha = 0.05$, Figs. 8b and S4), even though the morning peaks for CO is slightly higher during weekdays (Fig. 9b), which is consistent with An et al. (2015). Wang et al. (2013) found that NOx displays weekly cycle in the Beijing–Tianjin–Hebei metropolitan area, with higher level on weekdays than weekends. Qin et al. (2004)

observed a significant weekend effect in southern California, showing that in the morning traffic rush time, the concentrations of CO and $NO_2$ at weekends were about 18% and 37% lower than on weekdays. The difference between our study and other cities lies in the difference of fleet fuel structure, and the different weekly routine of human activities and the taxi driving trajectories (Xie et al., 2016).

    The median concentrations of CO and $NO_2$ during holidays are comparable to those in non-holidays, but are 18.3% lower

for $O_3$ (Fig. 8c). In addition, compared with the spatial distribution of $O_3$ concentration in holidays, we find that the concentrations of $O_3$ in Xinjiekou and its surrounding areas, where many shopping malls are located, are higher in non-holidays (Fig. S6). This may be related to the higher $NO_2$ concentrations in this area during holidays ($24.8 \pm 10.2$ $\mu g$ $m^{-3}$) than non-holidays ($20.6 \pm 4.82$ $\mu g$ $m^{-3}$). The hourly concentrations show no significant difference between holidays and non-holidays (Fig. 9c). The holidays include the periods of National Day (Oct. 1–7), the Spring Festival (Feb. 24–31),

Qingming Festival (Apr. 4–6), international labor day (May. 1–5), and the Dragon Boat Festival (Jun. 25–27). "Holiday effect" has been observed extensively for urban and regional air quality. For example, Xu et al. (2017) found that VOC tracers were significantly enhanced during the National Day holiday (from Oct. 1–10, 2014) in Yangtze River Delta (YRD) region, indicating that the "holiday effect" had a strong influence on the distribution and chemical reactivity of VOCs in the atmosphere. The reason why this effect is not observed in our study may be related to the relatively smaller sample size

during holidays. The sample size for holidays account for only 11.3 % of those for the non-holidays.

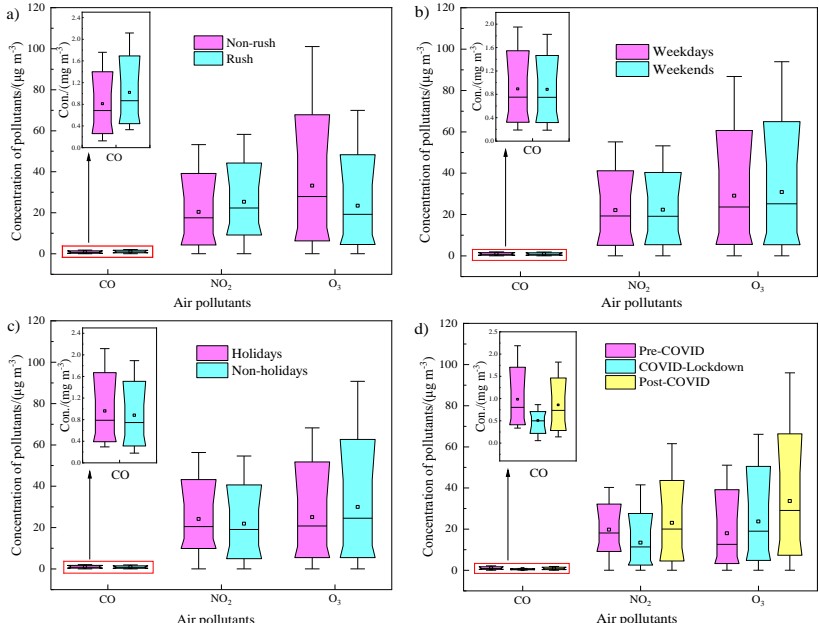

**Figure 8.** Variation of pollutants concentrations in rush/non-rush hours, weekdays/weekend days, holidays/non-holidays, and three stages of the COVID-19 pandemic. The dot in each box represents the mean value and the solid line represents the median value. Each box extends from the 25th to the 75th percentile. The whiskers (error bars) below and above the boxes represents the 10th and 90th percentiles.

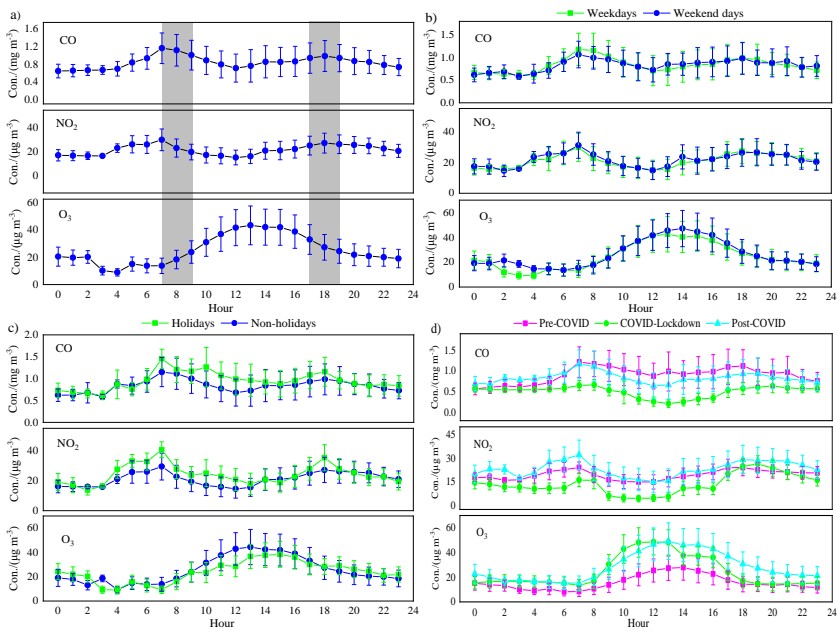

**Figure 9.** Diurnal cycles of three pollutants concentrations measured in rush/non-rush hours, weekdays/weekend days, holidays/non-holidays, and different stage of the COVID-19 pandemic by the taxi sensors. Error bars in panel a show the standard deviation of observations. Gray areas represent the rush hours, and the other represents the non-rush hours (a).

### 3.6 Traffic source contribution

Figs. 10a and 10b show the calculated contributions by traffic-related emission sources to the observed concentration of CO (referred to as contributions hereinafter). We find that the mean contribution calculated by BS method (42.6±11.5 %) is generally consistent with that obtained from PD algorithm (43.9±27.0 %). Their spatial patterns are also similar (Figs. 10a vs 10b). Although our data coverage is much larger than that of the Apte et al. (2017) study, we find that the reference method is still applicable in our research area. The contributions in highways, near tunnel entrances and exits (e.g. Jiuhuashan and

Xuanwuhu tunnel), railway station (Nanjing south station), and arterial roads (44–59 %) calculated by the both methods are higher than secondary roads, residential streets, and lowest in branch roads (29–39 %) (Table 3), which is consistent with the trend in traffic volumes. The patterns for NO₂ are quite similar to CO (Figs. S8c and S8d, Table 1), but the mean contribution

to $NO_2$ calculated by BS method (26.3±14.7 %) is lower than that obtained from PD algorithm (40.2±29.9 %). This difference is associated with the relatively higher uncertainty for $NO_2$ measurements by sensors (Sect. 2.2), while the results of PD method seem unaffected as the sensor bias are cancelled when calculating the difference between "peak" and "baseline" (Sect. 2.4).

Bottom-up emission inventory indicates that on-road transportation contributed ~11 % of total CO emissions from Nanjing in 2012 (Zhao et al., 2015). Considering the number of cars has increased ~80 % and the total CO emissions remained relatively stable (BSNM, 2019), the contribution of traffic sources in recent years is expected to be ~20 %. These values are much lower than what we calculated based on mobile monitoring data because of the lower spatial resolution of these regional inventories (e.g. 0.05° × 0.05°) (Zheng et al., 2014). They are unable to distinguish the emission characteristics of air pollutant within a street level (tens of meters), which leads to their underestimation of traffic-related emissions in the road micro-environment.

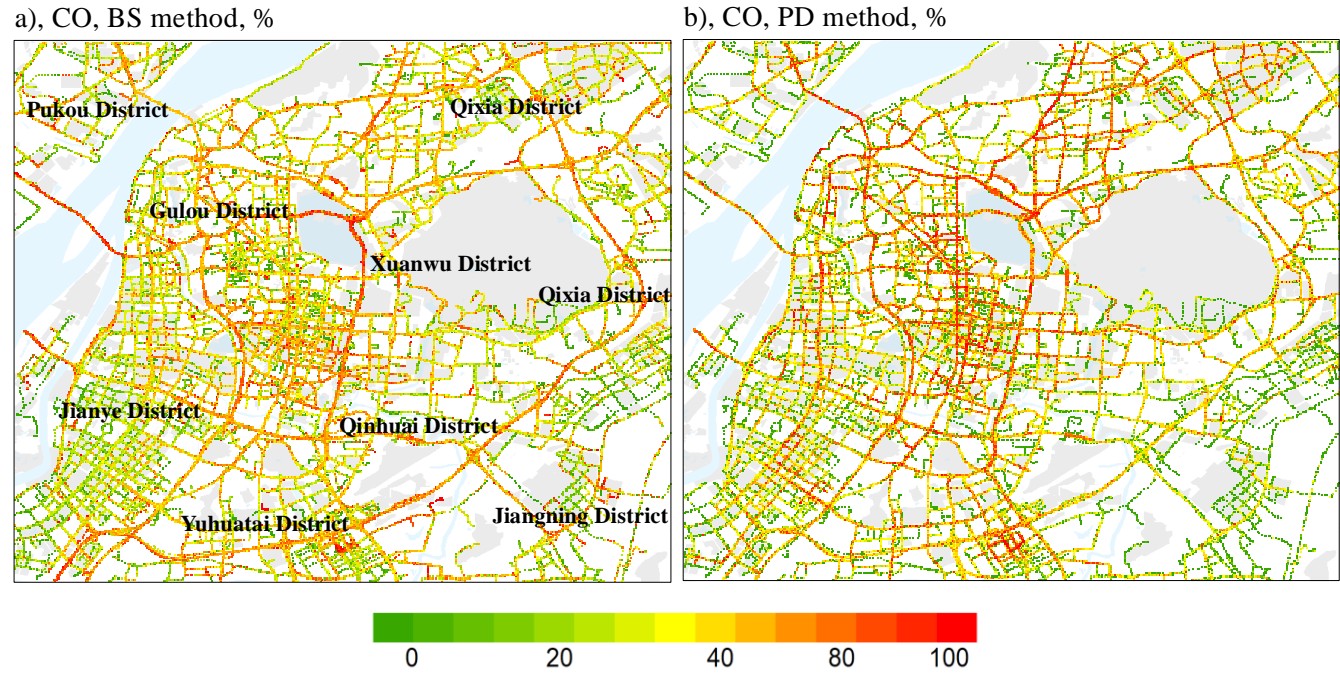

**Figure 10.** Contributions from traffic-related emissions calculated by stationary data method (a) and peak detection algorithm (b) for CO. © OpenStreetMap contributors 2019. Distributed under a Creative Commons BY-SA License.

**Table 3.** Contribution of traffic emissions to CO and $NO_2$ in different roads by two methods.

| Road types | Traffic emissions - CO, % | | Traffic emissions - $NO_2$, % | |
|---|---|---|---|---|
| | BS | PD | BS | PD |
| Highways | 48.3±10.4 | 51.0±20.4 | 32.5±14.5 | 41.4±22.5 |
| Arterials | 44.1±9.23 | 59.0±19.4 | 26.8±10.6 | 43.6±23.3 |
| Secondary | 40.2±11.7 | 47.6±23.9 | 22.8±13.2 | 35.2±25.1 |
| Residentials | 39.4±14.1 | 38.9±26.1 | 20.3±16.3 | 28.6±25.0 |
| Branch roads | 39.2±12.2 | 29.7±23.9 | 21.5±18.1 | 25.5±24.4 |

## 3.7 Impact of COVID-19 pandemic

Figs. 8d and 9d show the variation of air pollutant concentrations in different stages of the COVID-19 pandemic. The spatial distribution of concentrations and traffic contributions are also depicted in Figs. 11–12 and Figs. S9–S10. We divide the data into three stages: pre-COVID (P1, Oct. 1, 2019–Jan. 23, 2020), COVID-Lockdown (P2, Jan. 24–31, 2020 and Feb. 17–24, 2020), and post-COVID (P3, Mar. 1, 2020–Sep. 30, 2020). We find the median concentrations of CO and $NO_2$ were

the lowest in P2 (Fig. 9d). For example, the CO and NO$_2$ concentrations decreased by 44.9 % and 41.7 % from P1 to P2, respectively (Figs. 11 and S8). This pattern agrees well with the air quality station data over eastern China (Huang et al., 2020). We focus on the traffic sector as it is the most sensitive to lockdown measures, while other sectors, including power, industrial and residential sectors, remain relatively unchanged (Guevara et al., 2021). We find that from P1 to P2, the average traffic source contributions of CO and NO$_2$ by BS method decreased by 59.9 % and 51.8 %, respectively (Figs. 12 and S9). This is consistent with the transportation index data, which shows a 70 % reduction in eastern China cities during lockdown (Huang et al. 2020).

The observed CO and NO$_2$ concentrations recovered to a level similar to P1 during P3. The traffic-related source contributions were increased by 120 % and 131 % from P2 to P3 for CO and NO$_2$ (Figs. 11 and S9). Due to the limited data size and spatial coverage (only in some arterial roads and highways) during P2, the calculated contribution of traffic emissions to air pollutant may be not directly comparable to those shown in Fig. 9. But the changes of the contribution well track the change of traffic volume and human activities (Bao and Zhang, 2020). Our results also agree with top-down emission estimates from remote sensing data (Zhang et al. 2020), which showed the total NO$_2$ emissions decreased by 31–44 % from P1 to P2, but increased 67–85 % from P2 to P3.

The observed ozone concentrations show a different trend from other pollutants in the three stages. We find a pattern of P1 < P2 < P3 for O$_3$ median concentrations (Fig. 8d). The ozone concentration increased by 35.7 % from P1 to P2, and 48.7 % from P2 to P3 (Fig. S9). While the contribution of traffic emissions to ozone first decreased by 32.5 % from P1 to P2 period, and then increased by 39.3 % in P2 to P3 period (Fig. S10). This is firstly associated with the less titration of NOx during P2 as discussed earlier. In addition, the increased temperature and solar insolation in P2 and P3 also favor the photochemical formation of O$_3$ than in P1 (Xie et al., 2016; Fu et al., 2015; Reddy et al., 2010).

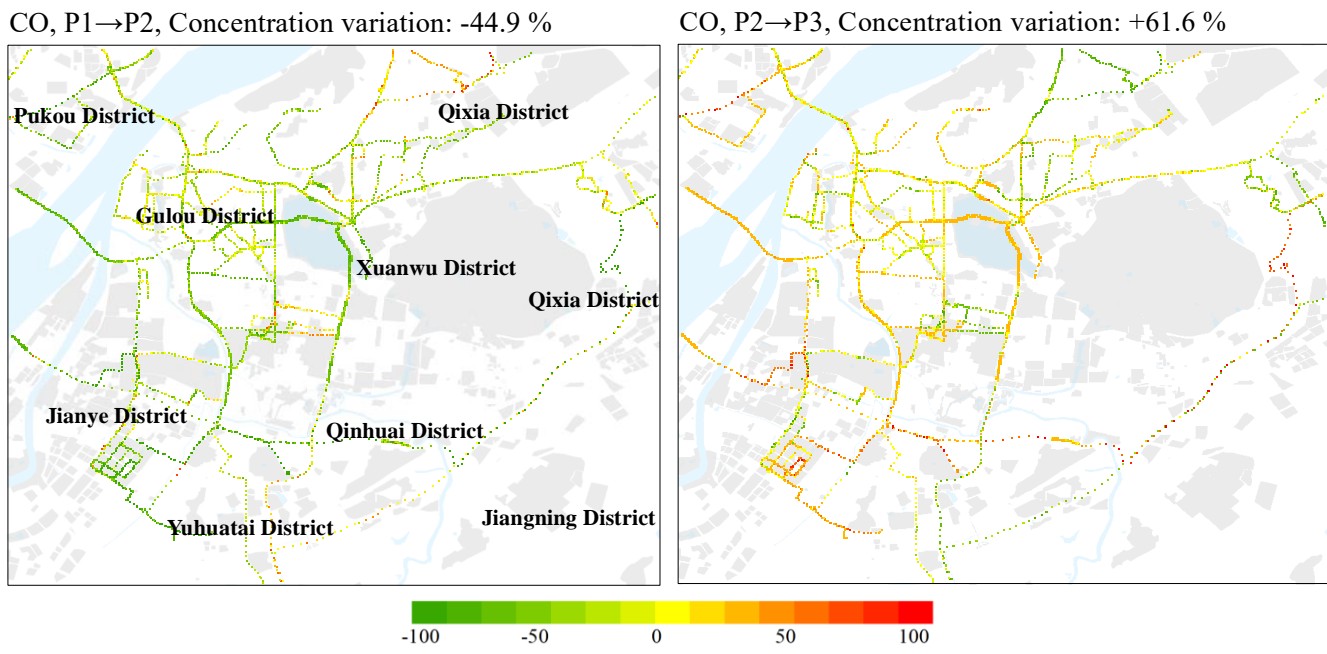

**Figure 11.** Changes of observed CO concentration in the three stages of the COVID-19 pandemic. P1, P2, and P3 are for pre-COVID, COVID-Lockdown, and post-COVID periods, respectively. © OpenStreetMap contributors 2019. Distributed under a Creative Commons BY-SA License.

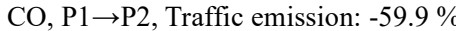

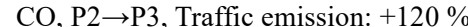

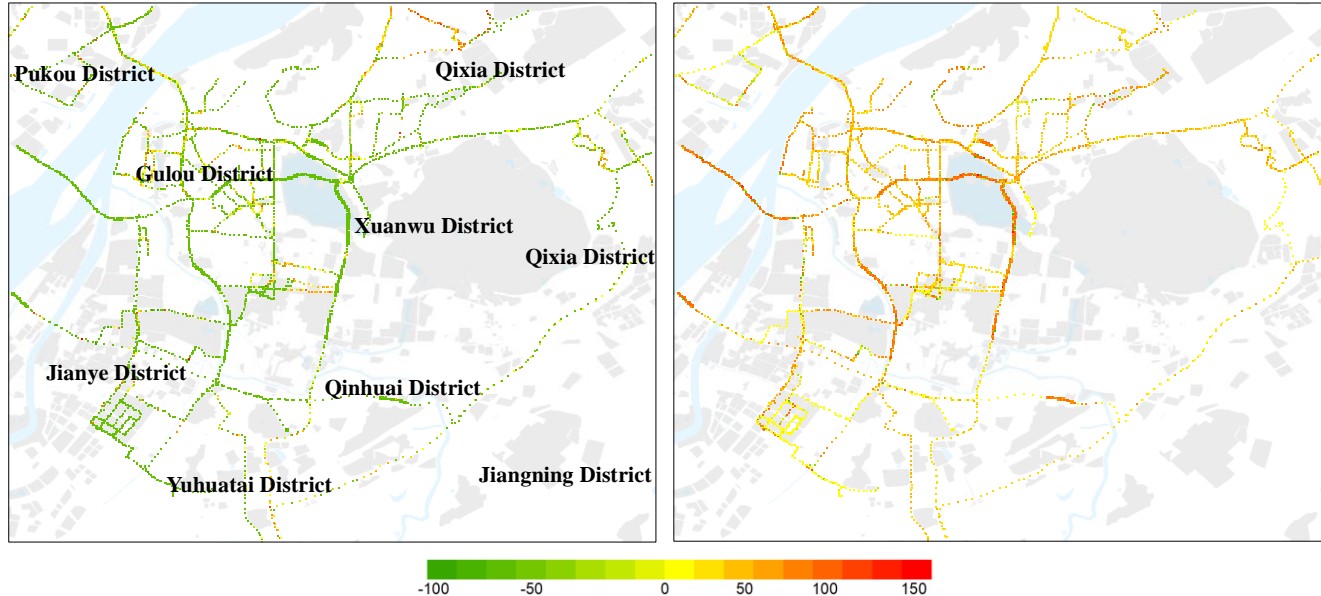

**Figure 12.** Changes of the contributions of traffic-related sources to CO in the three stages of the COVID-19 pandemic calculated by BS method. P1, P2, and P3 are for pre-COVID, COVID-Lockdown, and post-COVID periods, respectively. © OpenStreetMap contributors 2019. Distributed under a Creative Commons BY-SA License.

# 4 Conclusions

To accurately assess human exposure to urban air pollution requires a detailed understanding of the spatial and temporal patterns of air pollutant concentrations. Combined mobile monitoring with GIS technology, we obtained high-resolution (50 m × 50 m) spatial distribution maps of three air pollutants in the main urban area of Nanjing, which well demonstrates the spatial heterogeneity of pollutants at the micro-scales. We find that higher spatial resolution is useful to identify hotspots that are mainly affected by three types of air pollution emissions sources, namely, traffic, industrial, and cooking fumes. It also provides hints for air quality management and emission source control.

We calculate the contribution of traffic-related emissions to air pollutant in different grid points by combining mobile observation and station observation data. Compared with the peak detection method, the station data method is more reasonable for secondary pollutants as $O_3$, while the former is less affected by sensor bias. There are also some differences in the contribution of traffic emissions to air pollutants in different types of roads. Due to the impact of the COVID-19 pandemic, the mean concentrations of CO and $NO_2$ decreased by 44.9 % and 47.1 %, respectively, during the lockdown in Nanjing, and the contribution of traffic-related emissions also decreased by 59.9 % and 52.6 %. On the contrary, the concentration of $O_3$ increased by 35.7 %, respectively. After reopening, CO and $NO_2$ concentrations rebounded by 61.6 % and 48.2 %, and the contribution of traffic emissions both increased over 100 %, indicating the great impact of traffic emissions on urban air pollution.

*Data availability.* All validation data and data processing by GIS used in this work are accessed by contacting the authors.

*Author contribution.* YZ designed the research; SW performed the research; SW, YZ, ZW, and MY analyzed data; LW, XC, and AD provided validation data; MY, YL, and QL helped data analysis; MW, LZ, and YX provided monitoring instrument; SW and YZ wrote the paper.

*Competing interests.* The authors declare that they have no conflict of interest.

*Acknowledgments.* This study was supported by the National Key Research & Development Program of China (2016YFC0202000 and 2019YFA0606803), Jiangsu Innovative and Entrepreneurial Talents Plan, and the Collaborative Innovation Center of Climate Change, Jiangsu Province. The authors thank Rong Ye and Liang Luo for sample collection.

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
