# Peer review of "Mobile monitoring of urban air quality at high spatial resolution by low-cost sensors: Impacts of COVID-19 pandemic lockdown"

_Atmospheric Chemistry and Physics, 2020_

## Referee Comment (RC1) · Anonymous Referee #1 · 13 Dec 2020

This manuscript describes the deployment of low-cost air pollutant sensors for $O_3$, $NO_2$, and CO on taxis in Nanjing. This work is novel because it combines low-cost sensors with a distributed, quasi-random sampling platform. Overall the manuscript is appropriate for the journal, but it is not ready for publication at this time.

My main criticisms focus on the methods. As detailed in my comments below, the authors need to provide more information on the sensor package that they used. They do not even tell the readers whether these gas sensors were electrochemical, metal oxide, or something else. Additionally, the way that the data are assigned to points in space is not described in sufficient detail.

[Figure]

Line 26-27 It's unclear what is meant by global air pollution deteriorating by 8%. Is this for a specific pollutant?

Line 45 - you might need to capitalize Street View

Line 83 - What does SORPES stand for? Also, the link in this line returned a 404 error.

Line 92 suggests that the sensors were not calibrated until June 2020, however the measurements started in 2019. I am confused about the calibration schedule - hopefully the sensors were calibrated before the sampling on the taxis started. Please clarify.

I am not familiar with the XHAQSN-508. What kinds of sensors are these? Electrochemical? Metal oxide? More detail on the specific gas sensors is needed. Also, is the sample refreshed by pumping air pash the sensors, or do you rely on the airflow generated by the moving vehicle? If it's the latter, does it impact the performance to have the sensors stationary during calibration experiencing wind during sampling?

As shown in Figure 2, it seems that the calibration approach was to use the "forward" method - e.g., calibration models were built on one week of data, and then that calibration was used going forward. Other low-cost sensor studies use k-fold cross validation. In this approach, the data are divided into k chunks, and models are built on k-1 chunks and tested against the holdout. Does a k-fold cross validation of your data result in different (or perhaps better performing) calibration models?

The authors should specify what parameters were used in the calibration models. Is it just sensor raw signal, or are variables like temperature and humidity also included? Including a humidity term may improve performance of the NO2 model, as the authors note in lines 100-101 that the NO2 model may have a humidity bias.

Section 2.3 needs a better explanation of how the data are assigned to points in space. Data are logged every 10 seconds. Under many driving conditions (speed > 18 km/hr), the vehicle will cover more than 50 m in 10 sec. How is the resulting data assigned in

space? Is it the location of the vehicle when the data point is logged?

Is the final output the mean concentration in each grid? Since grid cells can be sampled unevenly across different days, other studies have first internally averaged the data by day. E.g., Apte et al 2017 compute the grid cell median for each sampling day, and then compute the mean of all daily medians.

How do large concentration spikes impact sensor performance? In our laboratory tests of electrochemical sensors, we observed that concentration spikes can cause the raw signal to remain high for several minutes. Presumably there are many spikes encountered during mobile sampling. Have the authors considered the potential impacts of these spikes? More broadly, have the authors considered that the sensors may not be able to reliably report at 10-sec resolution?

I'm confused by what is shown in Figure 4. I think that the standard error of the mean was calculated for each grid cell, and then averaged over all grid cells, but that is unclear. Were grid cells excluded if they did not meet a data threshold (e.g., if they did not have "enough" data)?

Figure 6 is hard to read. The lines indicating the roadways (or grid) are very thin, and it's hard to see the variation in the color scale with such thin lines.

Section 3.3 - This section is titled uncertainty analysis, but the discussion (especially lines 186-194) are more about spatiotemporal variability than uncertainty. This means that I am unclear on whether Fig 6 shows variability in measurement uncertainty (e.g., because of different sensitivities for different species), or if the variations in the coefficient of variation represent physical phenomena associated with emissions and chemistry.

Are the concentrations shown in Table 1 the mean concentration, or the concentration above background? The latter might be more informative.

Section 3.4.2 - how are the different types of roads defined?

[Figure]

Figure 9 shows diurnal patterns for the different pollutants. Were the data sub-selected in any way? I imagine that the locations sampled might be different across different times of day (e.g., maybe more time on highways at certain hours). It would be best if the data were somehow filtered - e.g., by only showing data collected on a certain road type, or by ensuring that data for each hour have a similar mix of road types sampled.

Figures 8 and 9 do not show a strong weekend effect. In the US, there is a strong weekend effect due to lower commercial diesel traffic (so there is lower NOx on weekends); but gasoline passenger cars have similar activity on weekends as weekdays, so CO is similar. Do your data suggest something about traffic patterns on weekdays versus weekends?

Figure 10, much like Figure 6, is hard to read. Maybe the authors could show a single panel in the main text and put the rest in the Supplement.

Line 293 - is the traffic percentage of O3 even a useful figure? As the authors note, attributing O3 is complicated because of secondary chemistry. I think they should remove the ozone estimate and focus here on CO and NO2.

Figures 11 and 12 are too faint to be readable.

---

## Referee Comment (RC2) · Anonymous Referee #2 · 22 Dec 2020

General Comments

This paper presents a mobile monitoring study of CO, NO2, and O3 concentrations in a major urban area. The research in this paper is a solid scientific study that adds to the knowledge we have of the variability in air concentrations in large urban areas. Below I detail some specific comments that should be addressed by the authors as well as some technical corrections.

Specific Comments

- Lines 85-90: please provide detailed information on the machine learning algorithm used, including the equations used to calibrate the data, what is considered a "sub-

stantial deviation" from the national network measurements, how recalibration was conducted if there was a substantial deviation, and how many times recalibration was needed.

- Lines 91-99: explain why you are using a machine learning algorithm. My understanding from your paper is that Figure 2a shows actual measurements, while Figure 2b shows the machine learning air concentration estimates for the mobile sensors compared to actual measurements at the fixed site. The correlations in Figure 2a are much better than those in Figure 2b, which would suggest that there is no need to train an algorithm to develop better estimates of concentrations. Why can't you simply use the measurements from the low-cost sensors for your calibration/validation? Is it because the study data were collected throughout the city, and not just near fixed monitors? If so, perhaps you can do a second calibration using data near fixed monitors, without the machine learning algorithm.

- Lines 128-130: this is a broad statement, and not true of all urban monitors. Can you provide citations to studies or reports that show that the stationary monitors do not have a significant impact from traffic emissions and are representative of urban background air quality?

- Lines 205-206 and Table 1: explain how you are identifying the main source contributions to the hot spots. Is it based on nearby sources and wind direction? Do different sources have different fingerprints (i.e., different relative concentrations of the measured pollutants)? Are there other studies showing that these sources had significant contributions at these locations?

- Lines 334-335: do the observations at fixed monitors support the theory that increased temperature/insolation is the cause of higher O3 concentrations in P3 as compared to P1?

Technical Corrections
**Interactive comment**

- Figure 2: both the x- and y- labels on the regression plots are labeled "station." Please change this to specify which station.

- Figure 5: the resolution isn't good on this figure. Can you re-plot with better resolution? Also, the yellow/orange colors are hard to differentiate in Figure 5b.

- Line 334: 'insulation' should be changed to 'insolation'

- Figure11: this figure is very hard to read. Can it be made a higher resolution or different color scheme?

---

## Short Comment (SC1) · 20 Jan 2021

D. Westerdahl

danewest03@gmail.com

General comments This paper reports on the findings of a year of monitoring conducted using a sensor pack on two taxis while they drove the streets of Nanjing, China. Based on the data reported the investigators developed concentration information plotted on the many roadways where data were collected. The opportunity to capture the impacts of major activity patterns associated with Covid-related restrictions is an interesting application of the results.

I concur with RC1 and will only add a few most concerning points beyond those in that review. The basic problem presented by this paper is that crucial methodolog-

ical/protocol descriptions regarding data collection activities are totally absent. I was unable to determine the nature of the sensor pack from the paper and a look at the Chinese company web site was not useful or clear-beyond the weight and size. There is no clear description of data adjustment which is mentioned. One this is very clear–the outcome of the monitoring data both before and after adjustment is startlingly good, beyond what most other users of sensors have reported. This enforces need to describe the process in detail. Overall, the uncertainty in monitoring and calibration practices makes it quite possible that the overall data set and interpretations might be impacted.

Specific comments Since the sensors are not described and data handling is also only somewhat described it appears possible that the data came from some sort of electrochemical cells. If this is true, it is quite possible that important Ozone/NO2 interactions occurred in, for example in the ozone data. This could have important implications on data observations and would show variable degrees of impacts depending on the mix of pollutants. The findings of fixed site calibration would differ from those made on road since the sensor experience a differing relative mix of NO2 and ozone.

Sensors seem perhaps only to be calibrated as study started and then once a month by comparison with an outdoor monitoring site (whether the sensors were tested in outdoor air or in some facility is unclear). Text in the first sentence of section 2.2 appears to state that sensor packs were placed at the campus supersite monthly for "at least seven days." No data are presented regarding the nature of the data at these monthly calibrations. Maintenance or data review are not described, however line 85 states "if the data deviated substantially from the nearest national network stations (shown as red stars in Figure 1), the instrument is also taken offline and re-calibrated." This statement indicates that there was some attention to reviewing the quality of data. This topic should be expanded and data on these calibration events should be included. The use of fixed site data should also be expanded. What was meant by the use of the "nearest" station in data review?

There is mention of a data calibration mentioned—"A supervised machine learning

methodology based on the Gradient Boost Decision Tree (GBDT) is used for data calibration" with a reference. But this should be fully described.

Line 187—consideration of observations vs. "life times" of each pollutant is incomplete and it is not clear how it applies to a near roadway urban environments where there is an impact of complex emissions/conversion and new emissions are present. This especially the case for the pollutant "NOx" mentioned on line 190—a pollutant that is not reported on in this study—the pollutant reported is NO2. The authors should provide a complete and careful consideration of these issues and they should be careful in the use of "NOx" vs the pollutant they measured. It seems to be used interchangeably in several places.

Figure 2—confidence In NO2 is not high seeing the good agreement with the fixed site dropped to R2=0.67. The authors suggest that this may be due to humidity impacts. NO2 and NO are probably the most important gaseous pollutant today in many urban near-roadway locations, but the authors have failed to follow up on the observations of possible poor model performance by repeating the calibration procedures. Further, for this pollutant, in this situation, it might be beneficial to see how the two sensor packs performed at each calibration. Current text only says they were in 'good agreement'. Authors should discuss the contributors to the mis match between agreement at cal vs validation for NO2. Is it clear that this fitting is successful as the sensors aged over the year?

What was the data capture completeness in this study? Were there any sensor replacements? Pollution observation examples would be helpful—provide specific time series examples.

Para beginning on line 205—where attribution of sources to observations is made. The actual basis for these is only general and not closely linked to the study. It appears to be conjecture.

Line 245—states that VOC control is necessary to control ozone at this site. This

Interactive
comment

may be true but is not studied or established by the investigators. It should be rewritten to reflect the basis for this statement.

The statement that lack of sunlight in the tunnel is the reason for low ozone may or may not be correct. A more complete consideration of emissions, ambient air ozone and reactions is called for here.

Conclusions Line 348—it is unclear that the following is established in this study—"We find that higher spatial resolutions are useful to identify hotspots that are mainly affected by five types of air pollution source emissions, namely, traffic, industrial, dust, and cooking fumes. It also provides hints for air quality management and emission source control." What assessments were made in this study to consider industrial dust, industrial fumes. . . ."?
* * *

---

## Author Comment (AC1) · 24 Jan 2021

This manuscript describes the deployment of low-cost air pollutant sensors for O3, NO2, and CO on taxis in Nanjing. This work is novel because it combines low-cost sensors with a distributed, quasi-random sampling platform. Overall the manuscript is appropriate for the journal, but it is not ready for publication at this time. My main criticisms focus on the methods. As detailed in my comments below, the authors need to provide more information on the sensor package that they used. They do not even tell the readers whether these gas sensors were electrochemical, metal oxide, or something else. Additionally, the way that the data are assigned to points in space is not

described in sufficient detail.

We are grateful to reviewer #1 for his/her effort reviewing our paper and his/her positive feedback. Here below we address the questions and suggestions raised by the reviewer #1. We provide more information on the sensor package used in this study and the way that the data are assigned to points in space.

1. Line 26-27 It's unclear what is meant by global air pollution deteriorating by 8%. Is this for a specific pollutant?

Re: The sentence in line 27-28 was modified as "The global urban air pollution (measured by PM10 or PM2.5) also deteriorated by 8%".

2. Line 45 - you might need to capitalize Street View

Re: We modified that as suggested.

3. Line 83 - What does SORPES stand for? Also, the link in this line returned a 404 error.

Re: We clarified this in line 94-96: "The instrument is placed at the Station for Observing Regional Processes of the EarthSystem (SORPES) in the Xianlin Campus of Nanjing University (https://as.nju.edu.cn/as_en/obsplatform/list.htm) for at least seven days before the taxi began sampling". Also, we replaced the link in line 96 with this: https://as.nju.edu.cn/as_en/obsplatform/list.htm.

4. Line 92 suggests that the sensors were not calibrated until June 2020, however the measurements started in 2019. I am confused about the calibration schedule - hope-fully the sensors were calibrated before the sampling on the taxis started. Please clarify.

Re: Thanks for your query. The XHAQSN-508 was calibrated once a month starting from September, 2019. The period June 1-17, 2019 was selected to do the calibration-validation (i.e. two-phase) experiment, but the one-phase calibration was conducted

every month. To clarify this, the sentence in line 94 was modified as: "The XHAQSN-508 is calibrated every month starting from September, 2019."

5. I am not familiar with the XHAQSN-508. What kinds of sensors are these? Electro-chemical? Metal oxide? More detail on the specific gas sensors is needed. Also, is the sample refreshed by pumping air push the sensors, or do you rely on the airflow generated by the moving vehicle? If it's the latter, does it impact the performance to have the sensors stationary during calibration experiencing wind during sampling?

Re: Thanks for pointing it out. The sentence in line 67-69 was modified as: "The instrument is equipped with internal gas sensors for CO, NO2, and O3 (dimensions: $290 \times 81 \times 55$ mm; weight: 1.0 kg) as well as two small in-built sensors for temperature and relative humidity, and is fixed in the top lamp support pole ($\sim$1.5 m above ground) of two Nanjing taxis (Figure 1)". And we also added the following sentences after that: "All three sensors are electrochemical-based sensors that can detect gaseous pollutants at levels as low as ppb (Maag et al., 2018). It is continuously powered by an external DC 12V power supply provided by a taxi battery. The sample is refreshed by pumping air to the sensors. There is an air inlet at the bottom of the instrument, which is also checked periodically to avoid blockage. Because it is fixed in the taxi top lamp, it can reduce the impact of different wind direction airflow". Then we added the relevant instrument description in line 75-77: "The monitoring data is automatically uploaded to a database in the cloud via the 4G telecommunications network. The monitoring system of CO, NO2, and O3 are configured to continuous measure at a frequency of once per 10 seconds, and their limit of detection (LOD) are 0.01 $\mu$mol/mol, 0.1 nmol/mol, and 0.1 nmol/mol, respectively".

6. As shown in Figure 2, it seems that the calibration approach was to use the "forward" method - e.g., calibration models were built on one week of data, and then that calibra-tion was used going forward. Other low-cost sensor studies use k-fold cross validation. In this approach, the data are divided into k chunks, and models are built on k-1 chunk sand tested against the holdout. Does a k-fold cross validation of your data result in

different (or perhaps better performing) calibration models?

Re: We clarify this by adding this sentence in line 101-103: "GBRT, an ensemble learning method, is a decision tree-based regression model that implements boosting to improve model performance using both parameter selection and k-fold cross validation".

7. The authors should specify what parameters were used in the calibration models. Is it just sensor raw signal, or are variables like temperature and humidity also included? Including a humidity term may improve performance of the NO2 model, as the authors note in lines 100-101 that the NO2 model may have a humidity bias.

Re: We acknowledged this point by adding this sentence in line 100-102: "GBRT needs to be trained by a dataset with target labels (Yang et al., 2017). It takes input variables including raw signals of sensors, other air pollutants concentrations, temperature and humidity. The stationary instrument data are taken as training targets". We also added a sentence in line 119-120: "To improve performance of the NO2 model, temperature and humidity are also involved in the training algorithm".

8. Section 2.3 needs a better explanation of how the data are assigned to points in space. Data are logged every 10 seconds. Under many driving conditions (speed > 18 km/hr), the vehicle will cover more than 50 m in 10 sec. How is the resulting data assigned in space? Is it the location of the vehicle when the data point is logged?

Re: We clarified this by adding some sentences in line 134-138: "The driving condition is highly variable and the taxi can travel more than 50 m in 10 seconds if the vehicle speed is over 18 km/hr. However, given the complexity of the driving conditions, we ignore the vehicle trajectory in the past 10 seconds but assign the measured values to the location of the vehicle at the time of data uploading. Then, combined with GIS technology, we calculate the average of all the data points over one year that fall in the same grid."

9. Is the final output the mean concentration in each grid? Since grid cells can be sampled unevenly across different days, other studies have first internally averaged the data by day. E.g., Apte et al 2017 compute the grid cell median for each sampling day, and then compute the mean of all daily medians.

Re: No, we used the direct average of all points throughout the year. By this mean, we treat all points in each grid equally. We clarified this by adding the following sentence in line 137-138: "Then, combined with GIS technology, we calculate the average of all the data points over one year that fall in the same grid." There are large minute-to-minute, hour-to-hour and day-to-day variabilities in pollutants concentrations. To calculate the mean (or median) of each day and then the mean of all daily mean (or median) is thus quite arbitrary. For example, why not using an eight-hour or weekly mean as the intermediate step? We argue that our method (i.e. direct mean of all points) is simpler and also robust if we have a large sample size.

10. How do large concentration spikes impact sensor performance? In our laboratory tests of electrochemical sensors, we observed that concentration spikes can cause the raw signal to remain high for several minutes. Presumably there are many spikes encountered during mobile sampling. Have the authors considered the potential impacts of these spikes? More broadly, have the authors considered that the sensors may not be able to reliably report at 10-sec resolution?

Re: Thanks for pointing it out. We indeed noticed the same phenomenon, and that is a drawback of our study. We acknowledge it by adding the following sentences in line 138-140: "One drawback of our study is the impact of spike concentrations on sensor performance. The sensors keep reporting high concentrations in an approximate one-minute period after exposure to large environmental concentration spikes. This effect would reduce the effective resolution of our gridded concentration map."

11. I'm confused by what is shown in Figure 4. I think that the standard error of the mean was calculated for each grid cell, and then averaged over all grid cells, but that
is unclear. Were grid cells excluded if they did not meet a data threshold (e.g., if they did not have "enough" data)?

Re: The review is correct and that's exactly what we did. We clarified it with relevant explanations in lines 176-177: "We calculate the standard error of the means of samples in each grid (SEM), and then averaged the SEM over all grid cells". We did not exclude any grid cells if they have more than two data points.

12. Figure 6 is hard to read. The lines indicating the roadways (or grid) are very thin, and it's hard to see the variation in the color scale with such thin lines.

Re: We have changed the image to a higher resolution, so we can see it clearly by zooming in. Very few readers read a paper version after all.

13. Section 3.3 - This section is titled uncertainty analysis, but the discussion (especially lines 186-194) are more about spatiotemporal variability than uncertainty. This means that I am unclear on whether Fig 6 shows variability in measurement uncertainty (e.g., because of different sensitivities for different species), or if the variations in the coefficient of variation represent physical phenomena associated with emissions and chemistry.

Re: Thanks for the suggestion. We modified the title of section 3.3 as "Variability analysis".

14. Are the concentrations shown in Table 1 the mean concentration, or the concentration above background? The latter might be more informative.

Re: We added a sentence to clarify it in line 233-234: "The pollutant concentrations shown in Table 1 are the values after deducting the background concentrations, which are calculated by the annual mean concentration of stationary stations".

15. Section 3.4.2 - how are the different types of roads defined?

Re: We clarified this in line 146-147: "We divide the urban roads in Nanjing area
into five types, including highways, arterial roads, secondary roads, branch roads, and residential streets (https://wiki.openstreetmap.org/wiki/Key:highway)".

16. Figure 9 shows diurnal patterns for the different pollutants. Were the data sub-selected in any way? I imagine that the locations sampled might be different across different times of day (e.g., maybe more time on highways at certain hours). It would be best if the data were somehow filtered - e.g., by only showing data collected on a certain road type, or by ensuring that data for each hour have a similar mix of road types sampled.

Re: We thank the reviewer for bringing this up. We acknowledged this point by adding this sentence in line 289-291: "The difference of the hourly variation of the mean sample of different types of roads over a year was small (Figure S6), so the data in Figure 9 is not filtered in anyway, but for each hour have a similar mix of road types sampled".

17. Figures 8 and 9 do not show a strong weekend effect. In the US, there is a strong weekend effect due to lower commercial diesel traffic (so there is lower NOx on weekends); but gasoline passenger cars have similar activity on weekends as weekdays, so CO is similar. Do your data suggest something about traffic patterns on weekdays versus weekends?

Re: Thanks for pointing it out. We added some discussion for this effect in line 300-305: "Wang et al. (2013) found that NOx displays weekly cycle in the Beijing–Tianjin–Hebei metropolitan area, with higher level on weekdays than weekends. Qin et al. (2004) observed a significant weekend effect in southern California, showing that in the morning traffic rush time, the concentrations of CO and NOx at weekends were about 18% and 37% lower than on weekdays. The difference between our study and other cities lies in the difference of fleet fuel structure, and the different weekly routine of human activities and the taxi driving trajectories (Xie et al., 2016)".

18. Figure 10, much like Figure 6, is hard to read. Maybe the authors could show a single panel in the main text and put the rest in the Supplement.

[Figure]

Re: We revised it as suggested.

19. Line 293 - is the traffic percentage of O3 even a useful figure? As the authors note, attributing O3 is complicated because of secondary chemistry. I think they should remove the ozone estimate and focus here on CO and NO2.

Re: Yes, it's a good suggestion. We have removed the ozone estimate in the revised paper.

20. Figures 11 and 12 are too faint to be readable.

Re: We have revised it in the revised paper.

---

## Author Comment (AC2) · 24 Jan 2021

General Comments This paper presents a mobile monitoring study of CO, NO2, and O3 concentrations in a major urban area. The research in this paper is a solid scientific study that adds to the knowledge we have of the variability in air concentrations in large urban areas. Below I detail some specific comments that should be addressed by the authors as well as some technical corrections.

We thank the reviewer for this comment and the helpful suggestion. We have carefully addressed the reviewer's concerns. Please see below our replies. We hope he/she is satisfied with our answers and the new (figure) we provided.

[Figure]

Specific Comments

1- Lines 85-90: please provide detailed information on the machine learning algorithm used, including the equations used to calibrate the data, what is considered a "substantial deviation" from the national network measurements, how recalibration was conducted if there was a substantial deviation, and how many times recalibration was needed.

Re: The detailed information on the machine learning algorithm was added in line 101-105: "GBRT, an ensemble learning method, is a decision tree-based regression model that implements boosting to improve model performance using both parameter selection and k-fold cross validation. GBRT needs to be trained by a dataset with target labels (Yang et al., 2017). It takes input variables including raw signals of sensors, other air pollutants concentrations, temperature and humidity. The stationary instrument data are taken as training targets". Since we did not calculate the "substantial deviation" from the national network measurements, we deleted it in the revised manuscript.

2- Lines 91-99: explain why you are using a machine learning algorithm. My understanding from your paper is that Figure 2a shows actual measurements, while Figure 2b shows the machine learning air concentration estimates for the mobile sensors compared to actual measurements at the fixed site. The correlations in Figure 2a are much better than those in Figure 2b, which would suggest that there is no need to train an algorithm to develop better estimates of concentrations. Why can't you simply use the measurements from the low-cost sensors for your calibration/validation? Is it because the study data were collected throughout the city, and not just near fixed monitors? If so, perhaps you can do a second calibration using data near fixed monitors, without the machine learning algorithm.

Re: To clarify this, we added this sentence in line 90-94: "Different from traditional instruments, low-cost sensors have some limitations, such as dynamic boundaries, nonlinear response, signal drift, environmental dependencies and low selectivity, so it

is important that calibration procedures are applied with respect to these limitations (Maag et al, 2018). The sensors are usually trained with co-located data collected by reference methods before being deployed to actual measuring campaigns (Kaivonen and Ngai, 2020; Chatzidiakou et al., 2019; Bossche et al., 2015)". We added a sentence in line 98-100 to further clarify: "Comparing different calibration models, we found that machine learning algorithm can improve sensor/monitor agreement with reference monitors, and many previous studies have used this method (Qin et al., 2020; Esposito et al., 2018; Vito et al., 2018)." We also added a sentence in line 107-109: "The success of supervised model training with target labels (i.e. co-located with SORPES, Figure 2a) does not guarantee for its predicting power for conditions without labels (i.e. on road or co-located with SORPES but not feeding the station data to the algorithm, Figure 2b)".

3- Lines 128-130: this is a broad statement, and not true of all urban monitors. Can you provide citations to studies or reports that show that the stationary monitors do not have a significant impact from traffic emissions and are representative of urban background air quality?

Re: We clarified this by adding the following sentences in line 155-159: "Seven state-operated air quality observation stations in Nanjing are selected in our research, including Maigaoqiao, Caochangmen, Shanxi Road, Zhonghuamen, Ruijin Road, Xuanwu Lake, and Olympic Sports Center (Zhao et a., 2015; Zou et al., 2017), which are far away from major roads and large point sources, so they are usually used as regional backgrounds in different functional areas (Zou et al., 2017; An et al., 2015). For example, Zou et al. (2017) chose the Olympic Center station (G, Figure 1) to get the background characteristics of CO and NO2 in Nanjing".

4- Lines 205-206 and Table 1: explain how you are identifying the main source contributions to the hot spots. Is it based on nearby sources and wind direction? Do different sources have different fingerprints (i.e., different relative concentrations of the measured pollutants)? Are there other studies showing that these sources had significant

contributions at these locations?

Re: We clarify this in line 235-238: "To identify the main sources contributing to these hotspots, we use the different relative concentrations of the measured pollutants (Zhao et al., 2015). We also use field information around hotspots area, such as the existence of subway stations, construction sites, factories, and restaurants nearby". Other studies had consistent results as stated in line 252-254: "Previous studies have also found that the air pollutants "hotspots" are associated with traffic-related emissions [e.g., heavy-duty diesel vehicles (Targino et al., 2016) and vehicle congestion (Gately et al., 2017)] and high-density urban areas (Li et al., 2018)."

5- Lines 334-335: do the observations at fixed monitors support the theory that increased temperature/insolation is the cause of higher O3 concentrations in P3 as compared to P1?

Re: Yes, they do. We added several references to support it in line 375: ".……(Xie et al., 2016; Fu et al., 2015; Reddy et al., 2010)".

Technical Corrections

1- Figure 2: both the x- and y- labels on the regression plots are labeled "station." Please change this to specify which station.

Re: The Figure has changed in the revised version. The x- and y- labels in Fig. 2a represents sensor-1 and sensor-2 respectively, while in Fig. 2b represents SORPES station and sensors data respectively.

2- Figure 5: the resolution isn't good on this figure. Can you re-plot with better resolution? Also, the yellow/orange colors are hard to differentiate in Figure 5b.

Re: We replace it with a high-resolution image, which can be viewed by zooming in.

3- Line 334: 'insulation' should be changed to 'insolation'

Re: We modified 'insulation' to 'insolation' in line 374.

4- Figure11: this figure is very hard to read. Can it be made a higher resolution or different color scheme?

Re: We have replaced it with a higher resolution image in revised version.

---

## Author Comment (AC3) · 5 Mar 2021

General comments This paper reports on the findings of a year of monitoring conducted using a sensor pack on two taxis while they drove the streets of Nanjing, China. Based on the data reported the investigators developed concentration information plotted on the many roadways where data were collected. The opportunity to capture the impacts of major activity patterns associated with Covid-related restrictions is an interesting application of the results.

Re: We appreciate Dr. Westerdahl for her/his effort to comment our manuscript and feedback. Dr. Westerdahl gives an accurate summary of our work and brings forward

constructive questions. We have addressed them below. We hope she/he is satisfied with our answers.

1. I concur with RC1 and will only add a few most concerning points beyond those in that review. The basic problem presented by this paper is that crucial methodological/protocol descriptions regarding data collection activities are totally absent. I was unable to determine the nature of the sensor pack from the paper and a look at the Chinese company website was not useful or clear-beyond the weight and size. There is no clear description of data adjustment which is mentioned. One this is very clear–the outcome of the monitoring data both before and after adjustment is startlingly good, beyond what most other users of sensors have reported. This enforces need to describe the process in detail. Overall, the uncertainty in monitoring and calibration practices makes it quite possible that the overall data set and interpretations might be impacted.

Re: Thanks for your query. We included more details for the methodology of data collection in the revised manuscript. Please refer to our response to the comments of the reviewers. For example: About the sensors, we added a note in line 67-69: ". . .. . .as well as two small in-built sensors for temperature and relative humidity. . .. . .". And we also added the following sentences to explain the nature of the sensors after that: "All three sensors are electrochemical-based sensors that can detect gaseous pollutants at levels as low as ppb (Maag et al., 2018). It is continuously powered by an external DC 12V power supply provided by a taxi battery". The sentences of "The monitoring data is automatically uploaded to a database in the cloud via the 4G telecommunications network. . .. . ., and their limit of detection (LOD) are 0.01 $\mu$mol/mol, 0.1 nmol/mol, and 0.1 nmol/mol, respectively" was also added in line 75-77.

The reason for the good outcome of the monitoring date both before and after adjustment is that the GBRT was selected for data calibration in this paper. And we added following sentence to describe the method in line 100-102: "Comparing different calibration models, we found that machine learning algorithm can improve sensor/monitor agreement with reference monitors, and many previous studies have used this method

(Qin et al., 2020; Esposito et al., 2018; Vito et al., 2018)". And sentences in line103-106: "GBRT, an ensemble learning method, is a decision tree-based regression model that implements boosting to improve model performance using both parameter selection and k-fold cross validation. GBRT needs to be trained by the dataset with target labels (Yang et al., 2017). It takes input variables including raw signals of sensors, other air pollutants concentrations, temperature and humidity. The stationary instrument data are taken as training targets".

2. Specific comments. Since the sensors are not described and data handling is also only somewhat described it appears possible that the data came from some sort of electro-chemical cells. If this is true, it is quite possible that important Ozone/NO2 interactions occurred in, for example in the ozone data. This could have important implications on data observations and would show variable degrees of impacts depending on the mix of pollutants. The findings of fixed site calibration would differ from those made on road since the sensor experience a differing relative mix of NO2 and ozone.

Re: We clarified this in line 69-70: "All three sensors are electrochemical-based sensors that can detect gaseous pollutants at levels as low as ppb (Maag et al., 2018)". And the following sentence was added in line 121-122: "Owing to the interaction between O3 and NO2, the detection accuracy of these two chemicals are influenced, especially for NO2 (Ivanovskaya et al., 2001)".

3. Sensors seem perhaps only to be calibrated as study started and then once a month by comparison with an outdoor monitoring site (whether the sensors were tested in outdoor air or in some facility is unclear).

Re: We clarified this by adding a sentence in line 92-94: "The sensors are usually trained with co-located data collected by reference methods before being deployed to actual measuring campaigns (Kaivonen and Ngai, 2020; Chatzidiakou et al., 2019; Bossche et al., 2015)". And the word "outdoor" was added in line 94-96: "The instrument is placed at the outdoor Station for Observing Regional Processes of the EarthSystem (SORPES) in the Xianlin Campus of Nanjing University (https://as.nju.edu.cn/as_en/obsplatform/list.htm) for at least seven days before the taxi began sampling".

Text in the first sentence of section 2.2 appears to state that sensor packs were placed at the campus supersite monthly for "at least seven days." No data are presented regarding the nature of the data at these monthly calibrations. Maintenance or data review are not described,

Re: We clarified this in line 97-99: "The collected data is calibrated against standard instruments (Thermo Fisher Scientific 48i, 42i, and 49i, USA for CO, NO2, and O3, respectively)". And we also added a sentence in line 99-100: "The instrument precision is $\pm$ 2ppbv for O3, and $\pm$ 1% and $\pm$ 4% for CO and NO2, respectively, which have been used in many other studies and found to perform well for long-term runs (Ding et al., 2013; Herrmann et al., 2013)".

However line 85 states "if the data deviated substantially from the nearest national network stations (shown as red stars in Figure 1), the instrument is also taken offline and re-calibrated." This statement indicates that there was some attention to reviewing the quality of data. This topic should be expanded and data on these calibration events should be included. The use of fixed site data should also be expanded. What was meant by the use of the "nearest" station in data review?

Re: We did not calculate the "substantial deviation" from the national network measurements, so we deleted this sentence in the revised manuscript.

4. There is mention of a data calibration mentioned about "A supervised machine learning methodology based on the Gradient Boost Decision Tree (GBDT) is used for data calibration" with a reference. But this should be fully described.

Re: We clarified this by adding the following sentences in line 103-106: "GBRT, an ensemble learning method, is a decision tree-based regression model that implements

boosting to improve model performance using both parameter selection and k-fold cross validation. GBRT needs to be trained by the dataset with target labels (Yang et al., 2017). It takes input variables including raw signals of sensors, other air pollutants concentrations, temperature and humidity. The stationary instrument data are taken as training targets".

5. Line 187 about consideration of observations vs. "life times" of each pollutant is incomplete and it is not clear how it applies to a near roadway urban environments where there is an impact of complex emissions/conversion and new emissions are present.

Re: To clarify this, we added the following sentences in line 227-229: "Lifetime (or residence time) is the average time for a chemical compound that is transported in the atmosphere before it is deposited or consumed by chemical reactions. It is associated with its spatial scale of variability. The longer the lifetime, the more uniform the concentrations are distributed".

This especially the case for the pollutant "NOx" mentioned on line 190 about a pollutant that is not reported on in this study about the pollutant reported is NO2. The authors should provide a complete and careful consideration of these issues and they should be careful in the use of "NOx" vs the pollutant they measured. It seems to be used interchangeably in several places.

Re: We measure NO2. But here we use NOx (=NO+NO2) as NO and NO2 are in a fast chemical cycle. It is thus more meaningful to use the lifetime of NOx instead of NO2. In other part of the manuscript, we also replace NOx with NO2 if we are referring to the chemicals we measure (e.g. line 260).

6. Figure 2 about confidence in NO2 is not high seeing the good agreement with the fixed site dropped to R2=0.67. The authors suggest that this may be due to humidity impacts. NO2 and NO are probably the most important gaseous pollutant today in many urban near-roadway locations, but the authors have failed to follow up on the

observations of possible poor model performance by repeating the calibration proce-
dures.

Re: We added a sentence to clarify this in line 121-122: "Owing to the interaction
between O3 and NO2, the detection accuracy of these two chemicals are influenced,
especially for NO2 (Ivanovskaya et al., 2001)".

Further, for this pollutant, in this situation, it might be beneficial to see how the two
sensor packs performed at each calibration.

Re: We tried so but there was no significant difference between the two sensors packs
performed at each calibration.

Current text only says they were in 'good agreement'. Authors should discuss the
contributors to the mismatch between agreement at cal vs validation for NO2. Is it
clear that this fitting is successful as the sensors aged over the year?

Re: We included more explanations for the mismatch between sensors and reference
method for NO2 (see responses to comments above). We further clarified the "aging"
issue by adding a sentence in line 123-126: "The accuracy of the sensor generally
decreases with time (aka aging) due to the evaporation of the electrolyte (Ribet et
al., 2018). However, we find no significant decrease in the R2 values for the three
pollutants during our campaign. It seems that the machine-learning algorithm could
successfully compensate the aging of the sensors."

7. What was the data capture completeness in this study?

Re: Figure 5 shows the completeness of data capture.

Were there any sensor re-placements?

Re: It happens a lot. We explained it in line 201-205: "As shown in Figure 5b, the
median number of repeated frequency in each grid is 66 (18, 286), with the highest
value of 15449 in Nanjing South Railway Station and the lowest in some residential

roads (1). The repeated frequencies in each 50-m grid along the arterial roads and Neihuan line are higher than other types of roads, i.e. Zhongyang road, Huju road, Neihuandong and Neihuanxi lines (Figure 5b)".

Pollution observation examples would be helpful about provide specific time series examples.

Re: That's a good suggestion. We added a sentence in line 206-208: "By comparing the time series of the air pollutant concentrations with that from nearby state-operated air quality stations (A' and E', with repeated frequencies > 500), we find that the results are consistent (Figure S1), which shows the stability and reliability of our data". And we also provide the original dataset for other researchers, so they can analyze the results in similar ways.

8. Para beginning on line 205 about where attribution of sources to observations is made. The actual basis for these is only general and not closely linked to the study. It appears to be conjecture.

Re: The observed concentrations are only a part of the basis for our source contribution. We also comprehensively analyze the pollution sources in hotspots through field surveys. We clarified this by adding a sentence in line 246-249: "To identify the main sources contributing to these hotspots, we use the different relative concentrations of the measured pollutants (Zhao et al., 2015). We also use field information around hotspots area, such as the existence of subway stations, construction sites, factories, and restaurants nearby".

9. Line 245 about states that VOC control is necessary to control ozone at this site. This may be true but is not studied or established by the investigators. It should be rewritten to reflect the basis for this statement.

Re: We agree with Dr. Westerdahl. So we deleted this sentence in the revised manuscript.

10. The statement that lack of sunlight in the tunnel is the reason for low ozone may or may not be correct. A more complete consideration of emissions, ambient air ozone and reactions is called for here.

Re: We thank Dr. Westerdahl for this suggestion. We modified the sentence in line 292-293 as: "The O3 concentrations are lowest in tunnels, which is associated with the weak sunlight in the tunnel (Awang et al., 2015)". We also added a sentence: "Furthermore, due to the unfavorable diffusion conditions in the tunnel, NO2 concentrations are accumulated to a relatively high levels ($40.7\pm29.7$ $\mu$g/m3), which titrates O3. The tunnel also blocks the replenish of surrounding O3-rich air, resulting in lower O3 concentrations than other roads (Kirchstetter et al., 1996)".

---

## Author Response (AR3)

**Report #1:**

Revised submission-- "Mobile monitoring of urban air quality at high spatial resolution by low-cost sensors: Impacts of COVID-19 pandemic lockdown"

The authors have considered responses to each of the points raised by the reviewers. However, there are key points at the base of earlier comments that are still quite unclear. These include: 1) specifics on the sensors used as basis for data source and 2) the implications of sensor choice, and 3) the implications of sensor calibration protocols on the data and its use in a traffic impacted environment.

Re: We thank the reviewer for the further comments. Please see our responses to these three points below.

A final, perhaps less important general concern is with the title of this paper. The title clearly shows this was a study of the impacts of COVID lockdowns. This event was a "nice" add-on, but it is not at all a major part of the effort. I suggest a title that better reflects the nature of the study. Re: Our observation covers different stages of the COVID-19 pandemic and also reflects the impact of the COVID-19 pandemic on the air pollutants variations. This is also one of the main objectives of our research, so we think it is suitable to keep the title as is.

The overall concern is that while the modelling and allocation of pollutants to roadways are interesting demonstration of mobile sensor-based monitoring it is unclear that actual and accurate pollutant data represents air quality on roadways. And while the calibration data from a Nanjing University campus show good performance it is not clear that this translates to accurate data from on road measurements. And without further information on sensors and the data they produced for this study it is not possible to evaluate or accept the modelling results.

Re: Please see our response below.

**Key points**

**1. Specifics on sensors**

The current version now adds a few words to identify the sensors as electrochemical cells, but there are many possible sources of cells of this type, and they differ. Please provide the make and models of the sensors. And fully describe any special characteristics that apply to the sensor calibration and validation in this study. For example, do they include any chemical filtration? Re: We checked with our instrument vendor, but they only released the internal model number of these sensors and kept the ultimate make and models in confidence. However, we think this is acceptable as we conducted regular calibration and validation processes and the results indicate the sensors worked reasonably well (Section 2.2).

We clarified this by adding the internal model numbers of the sensors in line 66-67: "The instrument is equipped with internal gas sensors for CO (model XH-CO-50-7), NO2 (XH-NO2-5AOF-7), and O3 (XH-O3-1-7) (dimensions:  $290 \times 81 \times 55$  mm; weight: 1.0 kg) ....".

and some sentences to describe the sensor calibration were added in line 94-99: "For example, environmental conditions are known to cause nonlinear behavior as well (Popoola et al., 2016). Due to aging and impurity effects over a long time, low-cost sensors are prone to signal drift and low sensitivity (Kizel et al., 2018). In addition, cross-sensitivities differ largely according to the ambient temperature and level of gas the sensor is being exposed to (Lösch et al., 2008). So, multi-parameter joint calibration training is utilized to reduce the interference issue between air pollutants, including air pollutants concentrations, temperature and relative humidity".

**2. Implications of sensor choice**

Electrochemical sensors are not fully chemically specific in response. This complicates their use in complex atmospheres—such as near roadways due to established interactions between ozone and nitrogen dioxide. Most "ozone" electrochemical sensors are "oxidant" sensors since they respond to oxidants (especially ozone and NO2). Data from them should not be simply viewed as ozone, as is the case in this study, since it is performed on NO2-rich roadways where ozone is likely to be disproportionately lower than in ambient air. NO2 electrochemical sensors may also have interference issues. Without clear identification of the sensors and a description of any features or data correction steps taken, it is not possible to understand the nature of data from either the "ozone" or NO2 sensors. Use of data from these sensors in subsequent plotting and modeling are open to considerable uncertainty. The reviewer is concerned of the use of sensor data as an off-the-shelf product without sufficient quality control and attention to details.

Re: We agree with the reviewer that the sensor data cannot be used as off-the-shelf product. We therefore have a regular sensor calibration and validation process during our study. We think this is an acceptable practice as the results indicate **the sensors work reasonably well (section 2.2)** and **the uncertainty is explicitly acknowledged**.

We acknowledged the cross-interference between NO2 and O3 in line 131-132:

"..., the interaction between  $O_3$  and  $NO_2$  may influence the detection accuracy of these two chemicals, especially for  $NO_2$  (Ivanovskaya et al., 2001)".

We acknowledged the uncertainty of our measurements in variability analysis in line 140-141: "Overall, the sensor results have substantial uncertainty compared to reference methods, we thus focus on the relative temporal and spatial distributions rather than the absolute concentrations".

The data collected while on road may be quite different than those from ambient community sites. This could complicate ozone reported. The authors should describe how ozone data was produced from the output of sensors and whether they considered interferences due to NO2. And depending on the NO2 sensor, they should address interferences that may occur for that sensor in the atmosphere under study. Because the accuracy of these two data streams is essential to the overall study, the details presented at adequate depth to inform the reader of what was done.

Re: We indeed considered the interferences between  $NO_2$  and  $O_3$ . A sentence was added to clarify how the data was produced in line 76-77:

"Pollutant concentration data is generated by different voltage values sensed by gas sensors, which is automatically uploaded to a database in the cloud via the 4G telecommunications network".

We also clarified this by adding a sentence in line 96-99:

"cross sensitivities differ largely according to the ambient temperature and level of gas the sensor

is being exposed to. So, multi-parameter joint calibration training was utilized to reduce the interference issue between air pollutants in our research, including air pollutants concentrations, temperature and relative humidity".

**3. Calibration issues**

Calibration is a key activity to assure good data from sensors. In this case the calibrations were performed using periodic co-location at an air monitoring station located at a campus of Nanjing University that may be distant from the urban center. This is possibly an acceptable approach, however some studies have shown the calibration results may not be transferrable especially crossing different concentration ranges or different environments. Expanded consideration of environmental conditions included in the calibration and how the change of environmental conditions can affect the are needed to help ensure credibility of data when sensors are deployed in the field. Again, back to sensor selection and data issues— The mix of ozone and NO2 are likely to be quite different between the roadways and the fixed site. Are there near-road ambient air monitoring sites in Nanjing that can show NO2, NO an O3 data levels that can be compared to the university site (which seems quite far from busy roadways)? How do they differ? How might these differences impact the utility of the calibrations performed?

It is important in sensor-based papers, that calibration as part of robust quality control and assurance is evident to ensure sufficient confidence for data interpretation or data fusion with modelling results. Overall, these topics are weakly considered in the manuscript.

Re: We thank the reviewer to bring this up. However, we do not have access to a regular sampling site near the road that uses reference method to measure these pollutants. All the national network sites were deliberately positioned far away from roads to represent a regional background.

We acknowledge this drawback in the revised manuscript in line 106-108:

"One drawback of our study is that the air pollutants concentrations observed at SORPES are lower than those observed in a road environment, which might cause issues for the calibration process".

One further point is identified that is separate from the above point regarding sensors and sensor data. That is with regards to the use of data from these two taxis to represent an entire urban area road grid. It seems that in many of the observed roadway segment data were collected perhaps once or twice over a one- or two-month interval while in other cases several observations may have occurred on a single day. The reviewer is not convinced this small data can be representative enough to be compared with modeling results to draw meaning conclusion. How might the nature of the temporal (even seasonal) nature of data quality impact the representation of the city roadways? Are two mobile monitors enough for a city the size of Nanjing?

Re: We indeed notice that the trajectories of the two taxis cannot fully cover the entire city, especially over less-populated regions. The re-visiting frequency of some grid points are quite low (Fig. 5b). However, most ( $\sim$  70 %) of the grid points have a sampling frequency > 50. It indicates that the dataset may be good enough to represent a long-term average, despite that more sensors are indeed needed if we want to pursue an air quality map with both high temporal and spatial resolutions.

We clarified the re-visiting frequency on the main roads in the revised manuscript (line 207-209): "A total of 1.32 million pieces of data were obtained during the observation period, which covers 66.4 % of the major roads in Nanjing in the sampling domain with a large repeat-visit frequency [median repetition = 61 (14 and 264 as the lower and upper quartiles, respectively, the same hereinafter)] (Fig. 5a)".

and a sentence to clarify the re-visiting frequency in each grid was in line 213-215: "As shown in Fig. 5b, the median number of repeated frequency in each grid is 66 (18, 286), with the highest value of 15449 in Nanjing South Railway Station and the lowest in some residential roads (1)".

**Detailed comments**

Line 77—states that taxis with natural gas and electric power were used. It is unclear how many taxis, but line 67 says that 2 were used in this study. If this is correct, may we assume this means one each?

Re: It was clarified in line 69-70:

"Two taxis fueled with electricity and liquefied natural gas (one each) are selected to reduce the impact of emissions from the sampling vehicles themselves".

Section 2.2, line 90—the authors list important characteristics to be considered with use of sensors. The list does not include specificity and interference issues or adequate details/citations on these matters. Overall, section 2.2 needs considerable editing and clarification. Calibration is a key point for this study.

Re: We clarified the interference issues by modifying the sentence in line 92-94:

"Different from traditional instruments, low-cost sensors have some limitations, such as nonlinear response, signal drift, environmental dependencies, low selectivity, and **cross-sensitivity**, so it is important that calibration procedures are applied with respect to these limitations (Maag et al, 2018; **Lösch et al., 2018**)."

And some sentences were also added in section 2.2 in line 94-99:

"For example, environmental conditions are known to cause nonlinear behavior as well (Popoola et al., 2016). Due to aging and impurity effects over a long time, low-cost sensors are prone to signal drift and low sensitivity (Kizel et al., 2018). In addition, cross sensitivities differ largely according to the ambient temperature and level of gas the sensor is being exposed to (Lösch et al., 2008). So, multi-parameter joint calibration training was utilized to reduce the interference issue between air pollutants in our research, including air pollutants concentrations, temperature and relative humidity".

this sentence in line 106-108:

"One drawback of our study is that the air pollutants concentrations observed at SORPES are lower than those observed in a road environment, which might cause issues for the calibration process."

these sentences in line 130-132:

"To improve performance of the  $NO_2$  model, temperature and relative humidity have also been involved in the training algorithm. However, the interaction between  $O_3$  and  $NO_2$  may influence the detection accuracy of these two chemicals, especially for  $NO_2$  (Ivanovskaya et al., 2001)."

**and a final sentence in this section:**

"Overall, the sensor results have substantial uncertainty compared to reference methods, we thus focus on the relative temporal and spatial distributions rather than the absolute concentrations."

It is stated that NO2 comparisons with the fixed station calibrations, which was only fair ( $R^2=0.67$  for sensor 2), were improved by training and inclusion of t and rh. But the resulting improvements are not shown in this section.

Re: Sorry for the confusion, but  $R^2=0.67$  is already the results with considering temperature and relative humidity. We clarified this by adding the following sentences in line 130-132:

"To improve performance of the  $NO_2$  model, temperature and relative humidity have also been involved in the training algorithm. However, the interaction between  $O_3$  and  $NO_2$  may influence the detection accuracy of these two chemicals, especially for  $NO_2$  (Ivanovskaya et al., 2001)."

Line 107—added text is unclear and perhaps is important. "The success of supervised model training with target labels (i.e. co-located with SORPES, Figure 2a) does not guarantee for its predicting power for conditions without labels (i.e. on road or co-located with SORPES but not feeding the station data to the algorithm, Figure 2b)". Perhaps this is related to key point 2. However, it should be rewritten and clarified. If it is related to point 2 then it should be considerably expanded.

Re: We had the following sentences following this one:

"We use a calibration-validation methodology to evaluate the performance of the calibrated sensors (Chatzidiakou et al., 2019). This method includes two phases: first, the sampler was calibrated against the SORPES station for 10 days (Jun. 1-10, 2020), and the sensor data were used for sensor algorithm training as above described (Fig. 2a); second, we continued to place the sampler in the station (Jun. 11-17, 2020). However, the sensor data are not used for calibration but directly fed in the algorithm trained in the first phase. The results are compared with the station data (i.e. validation phase, Fig. 2b)."

**We added a sentence in line 96-99:**

"... cross sensitivities differ largely according to the ambient temperature and level of gas the sensor is being exposed to (Lösch et al., 2008). So, multi-parameter joint calibration training was utilized to reduce the interference issue between air pollutants in our research, including air pollutants concentrations, temperature and relative humidity".

And other modifications were listed above in section 2.2 in point 2.

Line 140—it is unclear how the overall data reductions employed highly time and spatially resolved data and generated hourly average data. What was this data used for?

Re: The hourly average data is used to evaluate the diurnal cycles of air pollutants (Figure 9). We modified the sentence in line 157-158 as:

"Similarly, we calculate the hourly average concentrations by considering only the data sampled in the same hour of different days."

Line 143—what was the accuracy and data completeness for GPS in the urban areas. Others have found such data difficult to collect reliable complete files in urban canyon conditions.

Re: Our GPS works quite well except some minor spatial offsets. We clarified this by adding a sentence in line 82-83:

"A GPS device (U-blox, Switzerland) is utilized to record the spatial coordinates and the spatial offsets are corrected by Arcgis 10.2 software".

Line 163—"background" data are generated by finding the minimum value of all the stations in the Nanjing area. Is it clear that they represent the urban "background"? Overall, this text is unclear. Are any of these remote from the city or roadside stations? If so, how might these impact any determinations of background? Roadside station, for example might have very low ozone levels due to reaction between NOx and ozone. But this might not be urban "background". Please expand discussion of background assumptions.

Re: We made an explanation in the revised manuscript in line 172-176:

"Seven state-operated air quality observation stations in Nanjing are selected in our research, including Maigaoqiao, Caochangmen, Shanxi Road, Zhonghuamen, Ruijin Road, Xuanwu Lake, and Olympic Sports Center (Zhao et a., 2015; Zou et al., 2017), which are far away from major roads and large point sources, so they are usually used as regional backgrounds in different functional areas (Zou et al., 2017; An et al., 2015). For example, Zou et al. (2017) chose the Olympic Center station (G, Figure 1) to get the background characteristics of CO and NO2 in Nanjing".

Line 185—reference to the Apte study may or may not be valuable or applicable here. It was performed in a small area of a city—16km2 and employed considerable resampling of roadways over a prolonged period. Does it apply well in this much larger city/region?

Re: The results obtained by the Apte study were similar to that by the background site method (Figure 10). And a sentence to clarify this was added in line 358-359:

"Although our data coverage is much larger than that of the Apte et al. (2017) study, we find that the reference method is still applicable in our research area".

Line 239/246 and table 1—how were specific sources of pollutants, such as cooking identified, as source contributors to hotspots? Only one cooking establishment is actually reported as a hotspot contributor. Basically, a structured assessment of hotspots vs. sources is valuable, but this paper does not present any robust information on what allowed the identification of contributions beyond visual sightings or general proximity.

Re: We added a sentence to acknowledge this limitation in line 261-263:

"This method has substantial uncertainties to attribute the potential sources to these "hotspots", and further source-receptor relationship and detailed chemical component analyze are required to identify the exact emission sources."

Report #2:

The authors have addressed the majority of my comments from the first round of review. I have several additional minor comments below.

Line 75 - is the data reported every 10s the average of the previous 10s, or an instantaneous measurement? If it's the former, it seems like the spatial assignment should correspond to the median of each 10s sampling interval.

Re: It's an instantaneous measurement and be modified in line 79-80:

"An **instantaneous measurement** of CO, NO2, and O3 concentrations is configured to continuous measure at a frequency of once per 10s sampling interval".

Line 104 - please be specific about what concentrations of other pollutants are used in the calibration models.

Re: It was clarified in line 113-114:

"It takes input variables including raw signals of sensors, air pollutants concentrations (CO, NO2, and O3), temperature and relative humidity".

Figure 2 - (1) please clarify that the scatter plots show concentrations rather than raw signals; (2) it seems like the slopes of these scatter plots are equally as important as the  $R^2$ .

Re: It was modified as suggested and the slopes of these scatter plots were added in Figure 2.

Equation 1 seems to show a normalized traffic contribution, but the accompanying text suggests that the equation shows an absolute traffic contribution. Please clarify.

Re: The equation 1 shows the normalized traffic contribution of pollutants. We clarified this by adding a sentence in line 176-178:

"... the **normalized** contribution from traffic-related emissions can be obtained by differencing the mobile measurements and the stationary ones **to minimize the influence of daily meteorological variations in the urban air quality**, following Bossche et al. (2015)".

The ozone "hotspots" in fig 7 seem like cold spots because of NOx titration. Are there any true  $O_3$  hotspots (i.e., where  $O_3$  is higher than the surroundings)?

Re: Thank you for pointing this out. We modified this by identifying the true  $O_3$  hotspots with the area where the pollutant concentrations are 50 % higher than nearby grids. The results were shown in Fig 7, and some sentences was added in line 272-276:

"As shown in Figure 7, the higher O3 concentrations in these hotspots area are mainly caused by higher NOx and VOCs emissions from the heavy traffic (W, 46.8±27.4  $\mu$ g m-3, Xie et al., 2016; Ding et al., 2013), cooking emissions (Q, 38.5±26.0  $\mu$ g m-3), and ozone precursors from industrial emissions [e.g., K (47.1±36.5  $\mu$ g m-3) and J (37.6±25.8  $\mu$ g m-3)], such as VOCs. In addition, biogenic VOC emissions also have a significant impact on the formation of ozone [(U (40.4±18.3  $\mu$ g m-3)) and V (33.5±20.4  $\mu$ g m-3), Liu et al., 2018)]".